# Continuous in situ synthesis of a complete set of tRNAs sustains steady-state translation in a recombinant cell-free system

Fanjun Li [1,2], Amogh Kumar Baranwal [1,2] & Sebastian J. Maerkl [1] ✉

Construction of a self-regenerating biochemical system is critical for building a synthetic cell. An essential step in building a self-regenerative system is producing a complete set of tRNAs for translation, which remains a significant challenge. Here, we reconstitute a complete set of 21 in vitro transcribed tRNAs and optimize their abundance to improve protein yield. Next, we show that protein expression in the PURE transcription-translation system can be achieved by in situ transcribing tRNAs from 21 linear tRNA templates or a single plasmid template. To enable synthesis of mature tRNAs from a circular template, we employ either a nicked plasmid template or *T. maritima* tRNase Z to post-transcriptionally process the precursor tRNAs. We ultimately achieve continuous in situ synthesis of a complete set of tRNAs capable of supporting sustained, steady-state protein expression in PURE reactions running on microfluidic chemostats. Our findings advance the development of an autopoietic biochemical system.

Efforts are being made towards the bottom-up construction of a synthetic cell or biochemical system that exhibits the major hallmarks of life. Progress in this area will advance our understanding of fundamental processes in living cells and stimulate new biotechnology[1–3]. Self-regeneration, one of the most fundamental aspects of a living cell, is yet to be fully realized in an artificial system[4]. Among cell-free systems, the PURE system[5] is an ideal platform for the construction of a self-regenerative system, as PURE consists of defined and adjustable components, in comparison to complex lysate-based systems[6–8]. The first steps have been taken towards creating a self-regenerating PURE system by demonstrating that PURE is capable of self-regenerating some of its 36 non-ribosomal proteins[9,10]. Continuous self-regeneration of all 36 non-ribosomal proteins currently remains out of reach, but the use of continuous dialysis and microfluidic chemostats have been shown to increase protein production capacity of the PURE system[11], while optimizing PURE formulation enhanced protein synthesis efficiency[12]. Studies on encapsulating cell-free transcription and translation systems inside phospholipid vesicles and endogenously expressing channel proteins to allow selective permeability of nutrients, also significantly extended the reaction lifetime and

enhanced protein synthesis capacity[13]. Attempts have also been made to couple protein regeneration and DNA replication[14,15], synthesis of amino acids[16], and optimization of the energy regeneration process[17,18]. As part of the construction of a self-regenerative system, producing a complete set of transfer RNAs (tRNAs) directly in a cell-free system is essential, but has not yet been achieved[19].

tRNAs used in cell-free transcription-translation reactions are generally extracted from *E. coli*, where multiple isoacceptor tRNAs decode a single amino acid. However, 21 tRNAs are theoretically sufficient to decode formylmethionine and the 20 standard amino acids required for translation. In-situ production of tRNAs in PURE relies on in vitro transcription (IVT), where T7 RNA polymerase (T7 RNAP) is the most commonly used polymerase[20]. However, T7 RNAP also creates multiple by-products, including abortive products and transcripts with extra 3′ nucleotides[21–23]. This inhibits tRNA functionality, as they require a CCA sequence at the 3′-terminus for aminoacylation and ribosome interaction during protein synthesis[24,25]. DNA templates traditionally require an appropriate promoter sequence consisting of a T7 RNAP recruitment site and a purine initiation site for efficient transcription[26]. Unfortunately, several *E. coli* tRNAs are not purine-

[1]Institute of Bioengineering, School of Engineering, École Polytechnique Fédérale de Lausanne, Lausanne, Switzerland. [2]These authors contributed equally: Fanjun Li, Amogh Kumar Baranwal. ✉e-mail: sebastian.maerkl@epfl.ch

initiated[27]. To enhance the transcription efficiency of poorly transcribed tRNAs, an additional sequence with a strong initiation site, encoding either a self-cleaving ribozyme[28] or a cut site for RNase P[29–31] can be introduced upstream of the tRNA gene. Following enzymatic cleavage, mature tRNAs suitable for protein translation were generated. Previously, a set of 21 tRNAs, transcribed in vitro with the RNase P method and lacking any post-transcriptional modifications, was shown to support protein synthesis in PURE[31]. In the same work, the authors also reported rewriting the genetic code by assigning Ala to the Ser codon. More recently, Miyachi et al. demonstrated protein expression in the PURE system using 15 purine-initiated tRNAs transcribed in situ, but still required the exogenous addition of 6 chemically synthesized tRNAs[19]. The same work also demonstrated DNA replication of a tRNA^Ala plasmid template coupled with tRNA^Ala synthesis and protein expression in PURE. Despite these advances, in situ synthesis of a complete and functional set of 21 tRNAs in PURE has not yet been achieved.

In this study, we develop strategies for enabling in situ tRNA synthesis in the PURE cell-free system and achieve continuous in situ synthesis of all 21 tRNAs alongside stable protein production for up to 20 h. We first prepare 21 tRNAs through IVT and improve the transcription yields of four tRNAs by optimizing the tRNA sequence and the corresponding promoter. We verify functionality of the 21 IVT tRNAs by showing they enable protein synthesis in the PURE system, although giving rise to lower yields compared to those achieved with tRNA purified from *E. coli*. We show that adjusting the abundance of IVT tRNAs helps improve protein yield. Next, we demonstrate that protein expression can be achieved with in situ-synthesized tRNAs by using either 21 linear DNA templates or a single plasmid encoding all 21 tRNAs. To enable the synthesis of mature tRNAs with a circular template, we employ either Nt.BspQI to create a nicked template or *T. maritima* tRNase Z to process the precursor tRNAs, showcasing the potential of integrating a circular template for tRNA synthesis in a cell-free system. These and other improvements ultimately allow us to perform continuous, steady-state protein synthesis based on fully in situ transcribed tRNAs from either 21 linear DNA templates or a single plasmid DNA template in PURE reactions running on microfluidic chemostats. These results provide a step forward in the realization of a self-regenerating synthetic system.

## Results

### Preparation of 21 IVT tRNAs

Cell-free production of tRNAs through IVT typically employs T7 RNAP, a linear double-stranded DNA template, and NTPs. For T7 RNAP-based transcription, the DNA template must contain a promoter sequence consisting of a polymerase recruitment site and a purine initiation site[26]. Various T7 promoters have been previously identified[32] but, to our knowledge, only the T7 class III promoter $\phi$6.5 has been used to transcribe tRNAs, leading to efficient transcription only for guanine-initiated (G-initiated) tRNAs. For efficient transcription of adenine/uracil/cytosine-initiated (A/U/C-initiated) tRNAs, an extra sequence with a strong initiation site was introduced between the promoter sequence and the tRNA gene, which required removal after transcription by additional enzymatic reaction steps[28–31].

In this study, we attempted to IVT all 21 tRNAs without requiring additional 5′-processing enzymes. To begin, we utilized a set of linear DNA templates (ltDNAs) containing the commonly used T7 promoter $\phi$6.5 upstream of a wild-type tRNA gene (Fig. 1a, Supplementary Table 1, Supplementary Table 2) and performed separate run-off transcription reactions with T7 RNAP for each tRNA. Four non-G-initiated tRNAs, Asn, fMet, Ile, and Pro, transcribed at relatively low yields, while the other two non-G-initiated tRNAs, Gln and Trp, were transcribed with acceptable yields (Fig. 1b). We specifically attempted to improve transcription of the lowest-yield tRNAs: tRNA^Asn, tRNA^fMet, and tRNA^Ile. While G-initiated transcription by T7 RNAP under the T7

promoter $\phi$6.5 is widely used, Huang et al. demonstrated A-initiated transcription with the T7 class II promoter $\phi$2.5[33]. We tested the $\phi$2.5 promoter for transcription of the two A-initiated tRNAs, Ile (low yield) and Trp (medium yield). Using this specific promoter enhanced the transcription yields of tRNA^Ile and tRNA^Trp by 6-fold and 2-fold, respectively (Fig. 1c). A previous study used engineered tRNA^Asn (T-to-G substitution at position 1, A-to-C substitution at position 72) and tRNA^fMet (C-to-G substitution at position 1) for peptide synthesis and showed that the decoding fidelity was not affected by the mutations[30]. Therefore, we introduced the same substitutions to the DNA template for tRNA^Asn and tRNA^fMet, which we refer to as tRNA^fMet_mut and tRNA^Asn_mut, respectively. The transcription yields of tRNA^fMet_mut and tRNA^Asn_mut were enhanced by 56-fold and 33-fold, respectively (Fig. 1c). In summary, we obtained a complete set of 21 tRNAs at high yields through IVT, while eliminating the need for 5′ processing enzymes, by using two A-initiated tRNAs (Ile and Trp) transcribed with T7 promoter $\phi$2.5 and their corresponding wild-type gene, two engineered tRNAs (Asn and fMet) transcribed with T7 promoter $\phi$6.5 and their corresponding mutated genes and 16 tRNAs transcribed with T7 promoter $\phi$6.5 and their corresponding wild-type genes.

### Protein expression using 21 IVT tRNAs

We tested the activity of IVT tRNAs for protein expression in a PURE system by omitting exogenously added tRNAs (ΔtRNA PURE), and utilizing super folded GFP (sfGFP) as a reporter (Fig. 1d). The DNA template for sfGFP was designed according to a reduced genetic code (Supplementary Table 3, Supplementary Table 4) where each amino acid corresponds to only one codon as previously described[31]. We first prepared a mixture of 21 IVT tRNAs each at equal concentration, referred to as "uniform IVT tRNAs". We added the sfGFP template (4 nM) and various concentrations of uniform IVT tRNAs or *E. coli* tRNAs (NEB, E6840S) to the ΔtRNA PURE system and incubated reactions at 30 °C for 16 h while measuring fluorescence produced by translated sfGFP. The optimal sfGFP yield was obtained at around 0.5 µg/µl of uniform IVT tRNAs and gradually declined slightly at higher concentrations. In contrast, sfGFP yield using *E. coli* tRNAs reached an optimum at 1 µg/µl and remained constant at higher concentrations (Fig. 1e, f). The sfGFP yield with 0.5 µg/µl of uniform IVT tRNAs (17.9 ± 0.9 µg/ml) is ~7% of that achieved with the same concentration of *E. coli* tRNAs (267.3 ± 15.3 µg/ml). The lack of post-transcriptional modifications on IVT tRNAs likely reduced their translational activity, as Hibi et al. previously showed that introducing modifications on anticodon loop regions of some tRNAs partially restored activity[31].

Another factor contributing to the low efficiency of the IVT tRNAs could be that the uniform IVT tRNA abundance was suboptimal compared to *E. coli* tRNAs. Low levels of cognate tRNA may cause ribosome stalling at the codon, reducing the overall translation rate[34,35]. We therefore prepared a mixture of 21 IVT tRNAs matching *E. coli* tRNA abundance[36] (Fig. 1e), referred to as "weighted IVT tRNAs". Interestingly, sfGFP yield slightly increased as the concentration of weighted IVT tRNAs increased within our tested concentration range. The activity of weighted IVT tRNAs was lower than uniform IVT tRNAs at concentrations below 0.5 µg/µl but exceeded it above 1 µg/µl. The highest sfGFP yield obtained with weighted IVT tRNAs (35.6 ± 8.7 µg/ml) increased twofold compared to uniform IVT tRNAs and is ~12% of that achieved with *E. coli* tRNAs. Finally, we tested expression of mCherry, which was also successfully produced using the IVT tRNAs (Supplementary Fig. 1), further validating the activity of the 21 IVT tRNAs.

### Protein expression coupled with tRNA synthesis

Next, we attempted to couple protein translation with in situ tRNA transcription in the cell-free system. We prepared a mixture of 21 linear DNA templates (ltDNAs) at equal concentration of each ltDNA, referred to as "uniform ltDNAs". We assessed the activity of ltDNAs for protein

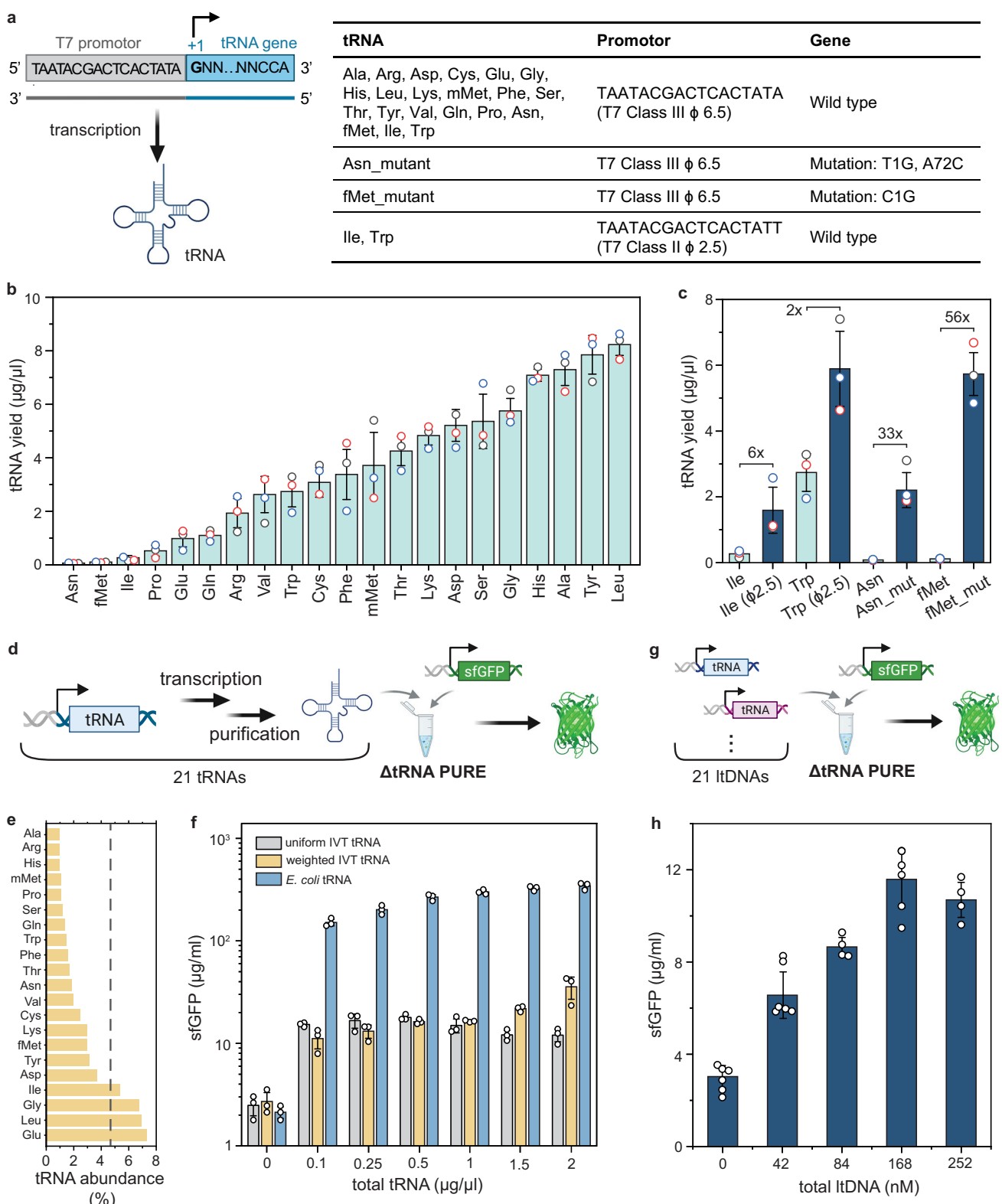

synthesis by incubating the uniform 21 ltDNAs and sfGFP template (4 nM) in a ΔtRNA PURE system at 30 °C for 16 h and measured fluorescence produced by translated sfGFP (Fig. 1g). At all tested concentrations of uniform ltDNAs, fluorescence levels higher than background were observed, suggesting that tRNAs were produced in situ within the ΔtRNA PURE system and were sufficiently functional to support sfGFP synthesis (Fig. 1h). The highest sfGFP yield obtained was 11.6 ± 1.2 µg/ml at a total concentration of 168 nM of the 21 ltDNAs,

which was ~65% and 39% of that achieved with 0.5 µg/µl of uniform and 1 µg/µl of weighted IVT tRNAs, respectively. Since the transcription efficiency of different tRNA genes varies (Fig. 1b), we anticipated that different amounts of tRNAs would be synthesized in the PURE system when the same ltDNA concentrations were used. The sfGFP expression might be limited by insufficient synthesis of certain tRNAs, such as tRNA$^{Pro}$, tRNA$^{Gln}$, and tRNA$^{Glu}$. Further optimizing the concentrations of individual ltDNAs might help improve protein synthesis in the future.

**Fig. 1 | Preparation of IVT tRNAs, and consequent protein synthesis using IVT recombinant tRNAs or in situ synthesized tRNAs. a** A schematic of in vitro transcription and a table summarizing the tRNA DNA templates used in this study. Created in BioRender. Maerkl, S. (2025) https://BioRender.com/eh7ck4h. **b** Transcription yield of tRNAs using a linear template containing a T7 class III promoter $\phi$6.5 upstream of the wild type tRNA gene ($n = 3$ replicates). **c** Comparison of tRNA yield before and after template optimization ($n = 3$ replicates). **d** A schematic describing the approach to test the activity of 21 IVT tRNAs in a ΔtRNA PURE system using sfGFP as a reporter. Created in BioRender. Maerkl, S. (2025) https://BioRender.com/eh7ck4h. **e** The abundance of *E. coli* tRNAs (yellow

bars) and the uniform tRNA concentration used (gray dotted line). **f** Expression of sfGFP using the indicated tRNA source in ΔtRNA PURE system ($n = 3$ replicates). **g** Schematic of testing the activity of in situ transcribed tRNAs from linear tRNA templates in ΔtRNA PURE system using sfGFP as a reporter ($n = 3$ replicates). Created in BioRender. Maerkl, S. (2025) https://BioRender.com/eh7ck4h. **h** The expression of sfGFP with various input concentrations of linear DNA templates for in situ transcription of tRNAs in the ΔtRNA PURE system ($n = 6$ for 0 and 42 nM ltDNA, $n = 4$ for 84 and 252 nM ltDNA, $n = 5$ for 168 nM ltDNA). Each dot represents a data point from an independent replicate. Bars and error bars represent the mean and standard deviation. Source data are provided as a Source Data file.

Importantly, we were able to achieve successful in situ transcription of all 21 tRNAs in a cell-free system.

## Protein expression from a single plasmid encoding all 21 tRNA genes

A circular DNA template encoding all tRNA genes would be advantageous for DNA replication and developing an artificial self-regenerating system[37–39]. Therefore, we investigated the feasibility of expressing all 21 tRNAs from a single circular plasmid template. An important challenge to overcome here is obtaining tRNAs with homogeneous 3′-CCA termini, which is required for aminoacylation and interaction with ribosomes[24]. Miyachi et al. previously solved this problem for a single tRNA by inserting a Nt.BspQI recognition site downstream of the tRNA$^{Ala}$ gene and using the Nt.BspQI-nicked plasmid as the transcription template for tRNA$^{Ala}$[19]. The Nt.BspQI treatment created a nick on one strand of the circular DNA for transcription of tRNA with a homogeneous 3′-CCA end, while keeping the other strand intact for DNA amplification.

Adapting this strategy, we inserted all 21 tRNA genes into a pUC19 vector, placing a T7 promoter upstream and an Nt.BspQI recognition site immediately downstream of each tRNA gene (Supplementary Fig. 2). The resulting construct was designed so that the Nt.BspQI-treated pUC19_21 tRNA genes could serve directly as a transcription template for producing 21 mature tRNAs (Fig. 2a). The Nt.BspQI-treated and untreated pUC19_21 plasmids are henceforth referred to as nicked and un-nicked plasmid, respectively. The transcribed products from nicked and un-nicked plasmid, were purified and analyzed with UREA-PAGE. The band profile of tRNA$_{nicked}$ at 50–100 nt largely resembled that of *E. coli* tRNA, suggesting that functional tRNAs were transcribed, albeit with a significant portion of undesired precursor tRNAs (pre-tRNAs) (Fig. 2b). We evaluated the activity of tRNAs transcribed and purified from either nicked or un-nicked plasmid, adding them to a ΔtRNA PURE reaction along with sfGFP template (4 nM), incubated the reaction at 30 °C for 16 h and measured fluorescence produced by sfGFP (Fig. 2c). The addition of 0.5 µg/µl and 1 µg/µl tRNA produced from the nicked plasmid resulted in a 3–4 fold higher sfGFP expression yield compared to the negative control, whereas adding the same amounts of tRNA produced from un-nicked plasmid had little effect, indicating that the tRNA produced from nicked plasmid produced functional tRNAs.

We then investigated whether tRNA can be transcribed from a nicked plasmid in situ within the PURE system by directly adding the purified nicked plasmid into the ΔtRNA PURE system containing 4 nM sfGFP template. At all tested concentrations of nicked plasmid, fluorescence levels higher than background were observed, suggesting that active tRNAs were produced in situ within the ΔtRNA PURE system to support sfGFP synthesis (Fig. 2d). Upon comparing the efficiency of ltDNAs with nicked plasmid at equal total gene concentrations (168 nM total ltDNA concentration is equivalent to 8 nM nicked plasmid), we found that uniform ltDNAs have higher activity than the nicked plasmid. This is not unexpected, as the UREA-PAGE analysis revealed a significant portion of undesired pre-tRNAs being transcribed from the nicked plasmid which could be due to incomplete nicking by Nt.BspQI.

## Use of *T. maritima* tRNase Z for post-transcriptional cleavage of pre-tRNAs

Although the use of a nicked plasmid for in situ transcription of tRNAs was successful, it is not necessarily ideal due to the requirement of generating a nicked plasmid template, which could lead to issues in coupling tRNA transcription and DNA replication at a later point in time. One possible strategy for 3′-end maturation involves incorporating a self-cleaving ribozyme at the 3′-end of the tRNA, which generates a tRNA bearing a 2′,3′-cyclic phosphate. This intermediate can then be converted to a mature 3′-OH end through enzymatic dephosphorylation, enabling aminoacylation and participation in translation. Coincidentally, while we were preparing our manuscript, this method was experimentally validated by another group[40]. To streamline 3′-end processing in vitro, we aimed to develop a single-enzyme solution that performs the complete maturation. We explored the use of a tRNA 3′ processing enzyme, *T. maritima* tRNase Z, to facilitate the removal of the 3′-trailer sequence and production of mature tRNAs from *E. coli* pre-tRNAs. In *E. coli*, the longer 3′-trailers on pre-tRNAs are first cleaved by the endonuclease RNase E before the final 3′ maturation step is performed by exonucleases (primarily RNase T and PH)[41]. By contrast, a single enzyme, tRNase Z, is responsible for the 3′ maturation process in the bacterium *Thermotoga maritima*, which cleaves the CCA-containing pre-tRNAs to the CCA triplet, yielding tRNAs ready for aminoacylation[42]. Therefore, we tested the activity of *T. maritima* tRNase Z against *E. coli* pre-tRNAs. A ltDNA coding for pre-tRNA$^{Ser}$ was designed by adding 50 random nucleotides, excluding any CCA triplets, downstream of the 3′ CCA end of the *E. coli* tRNA$^{Ser}$ gene (Supplementary Table 1). The IVT pre-tRNA$^{Ser}$ was used as a substrate and incubated with tRNase Z (Fig. 3a). UREA-PAGE analysis of the cleaved product revealed a dominant band of similar molecular weight as a run-off transcribed tRNA$^{Ser}$, suggesting *T. maritima* tRNase Z is active and produces properly sized tRNA$^{Ser}$.

We then tested whether tRNase Z could cleave pre-tRNAs transcribed from the un-nicked pUC19_21 tRNA gene plasmid followed by incubation with *T. maritima* tRNase Z. It should be noted that pUC19_21 does not include T7 terminators after each tRNA gene, and therefore produces a plethora of different concatenated tRNA transcripts. Upon treatment with tRNase Z, a smeared band was observed on a denaturing urea polyacrylamide gel, suggesting incomplete cleavage (Fig. 3b). Nevertheless, we purified the tRNase Z-treated pre-tRNAs and evaluated their activity in a ΔtRNA PURE* system containing 4 nM of sfGFP template. We also note that the commercial energy solution in the ΔtRNA PURE system was replaced with a laboratory-made energy solution in the ΔtRNA PURE* system as lower protein synthesis was observed with the commercial energy solution (Supplementary Fig. 3). At 1 µg/µl and 1.5 µg/µl of tRNase Z-treated pre-tRNAs, sfGFP expression yield were around 1.5 times higher than the groups with untreated pre-tRNAs, suggesting that *T. maritima* tRNase Z can use *E. coli* pre-tRNAs as substrates in vitro and produces functional tRNA for sfGFP synthesis, albeit with low efficiency (Fig. 3c). Interestingly, a significant increase in activity of untreated pre-tRNAs from 1 to 1.5 µg/µl was observed, which could be due to the residual endogenous tRNA processing enzymes in the PURE system, or due to increased levels of short tRNA segments produced by T7 RNAP during abortive cycles that

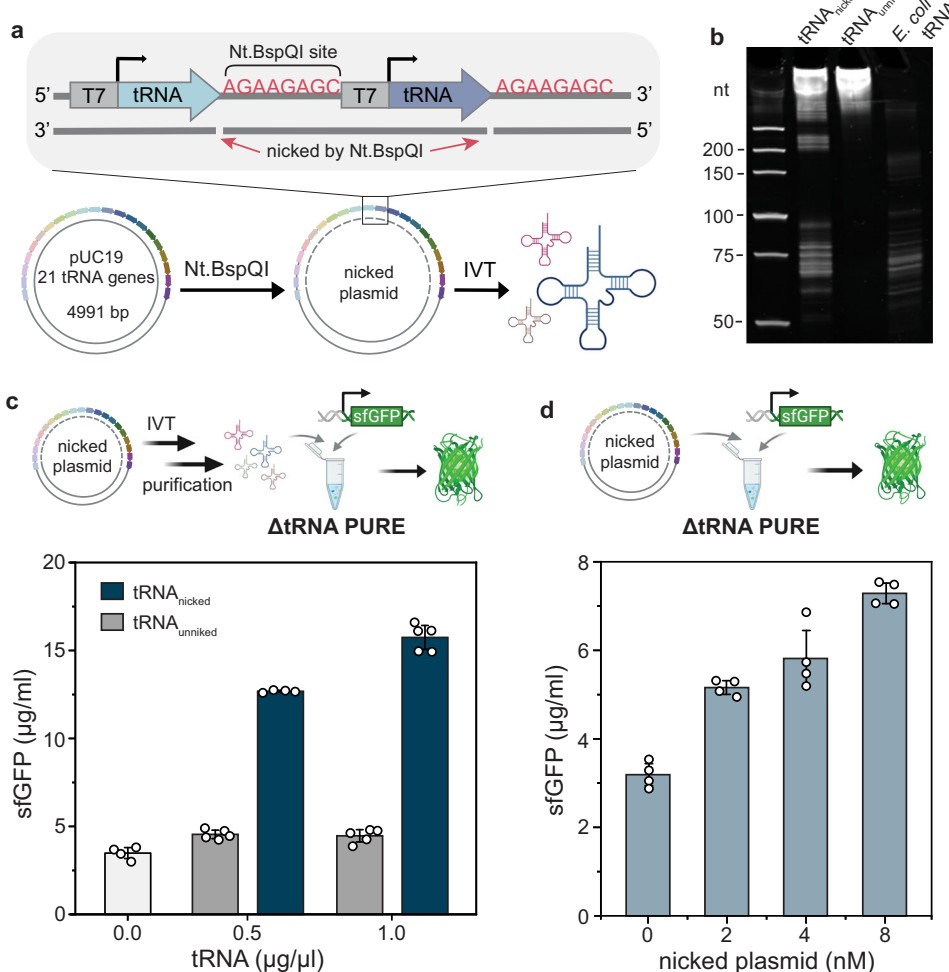

**Fig. 2 | Protein expression with a single plasmid encoding all 21 tRNA genes. a** A schematic describing the preparation of mature tRNAs from a circular template nicked with Nt.BspQI. Created in BioRender. Maerkl, S. (2025) https://BioRender.com/eh7ck4h. **b** UREA-PAGE analysis of tRNAs transcribed from nicked and un-nicked plasmid. *E. coli* tRNAs are also shown for comparison. **c** Schematic describing the approach to test the activity of IVT tRNAs transcribed from a nicked plasmid in ΔtRNA PURE system ($n = 4$ for 0 and 0.5 µg/µl tRNA$_{nicked}$, $n = 5$ for 0.5 and 1 µg/µl tRNA$_{unnicked}$, 1 µg/µl tRNA$_{nicked}$). The bar plot shows the expression of sfGFP with various concentrations of IVT tRNA added to the ΔtRNA PURE system.

Created in BioRender. Maerkl, S. (2025) https://BioRender.com/eh7ck4h. **d** Approach for testing the activity of nicked plasmid added directly to theΔtRNA PURE system for in situ tRNA transcription ($n = 4$). Created in BioRender. Maerkl, S. (2025) https://BioRender.com/eh7ck4h. The bar plot shows the expression of sfGFP with the indicated concentration of nicked plasmid added to the ΔtRNA PURE system. Each dot represents a data point from an independent replicate. Bars and error bars represent the mean and standard deviation. Source data are provided as a Source Data file.

happen to be mature and functional tRNAs. To improve the maturation process, we tested another circular plasmid (pETA_21 tRNA genes; Supplementary Fig. 4) where a wild-type T7 terminator was inserted downstream of each tRNA gene. Transcripts from this plasmid were incubated with tRNase Z and subsequently tested for activity in the ΔtRNA PURE system, but no significant improvement was observed (Supplementary Fig. 5). This could be due to the insufficient termination efficiency of the wild-type T7 terminator, resulting in the persistence of long transcripts. To address this, stronger T7 terminators[43] could be evaluated next. Besides, shuffling the positions of tRNA genes[40] may offer further enhancements.

**Continuous, steady-state in situ tRNA transcription and protein synthesis in a microfluidic chemostat**

Having successfully demonstrated that DNA templates could be used to synthesize functional tRNAs in situ in cell-free batch reactions, we sought to test whether it is possible to perform continuous, long-term tRNA transcription and protein synthesis in chemostat reactions and achieve steady-state conditions[44]. We used a microfluidic chemostat

device, similar to one previously described by Lavickova et al.[9]. Each microfluidic chip contains eight chemostat rings, all with a volume of 15 nl and fluidically hard-coded dilution fractions defined by reactor geometry (Fig. 4a). To load the PURE components into the rings, we combined tRNAs or tRNA templates, sfGFP template and energy components into a single solution (DNA/energy mix) and introduce it using one microfluidic inlet, while the protein/ribosome components are introduced using a separate inlet to prevent any reactions from occurring prior to mixing on-chip (Fig. 4b). mScarlet was added as a tracer to the protein/ribosome mix for validating proper chemostat function. We implemented a dilution step to replenish the energy source and enzymatic machinery while removing byproducts. 20% of the reactor volume was replaced every 20 min with a 3:2 ratio of DNA/energy mix:protein/ribosome mix. Each experimental condition was performed in duplicate in two rings and run for up to 20 h.

First, we verified the functionality of IVT tRNAs transcribed from 21 ltDNAs for on-chip steady-state protein synthesis (Fig. 4d). To set up the reactions, 0.5 µg/µl uniform IVT tRNAs were added to the DNA/energy mix. The positive control rings contained 1.5 µg/µl *E. coli* tRNAs (NEB,

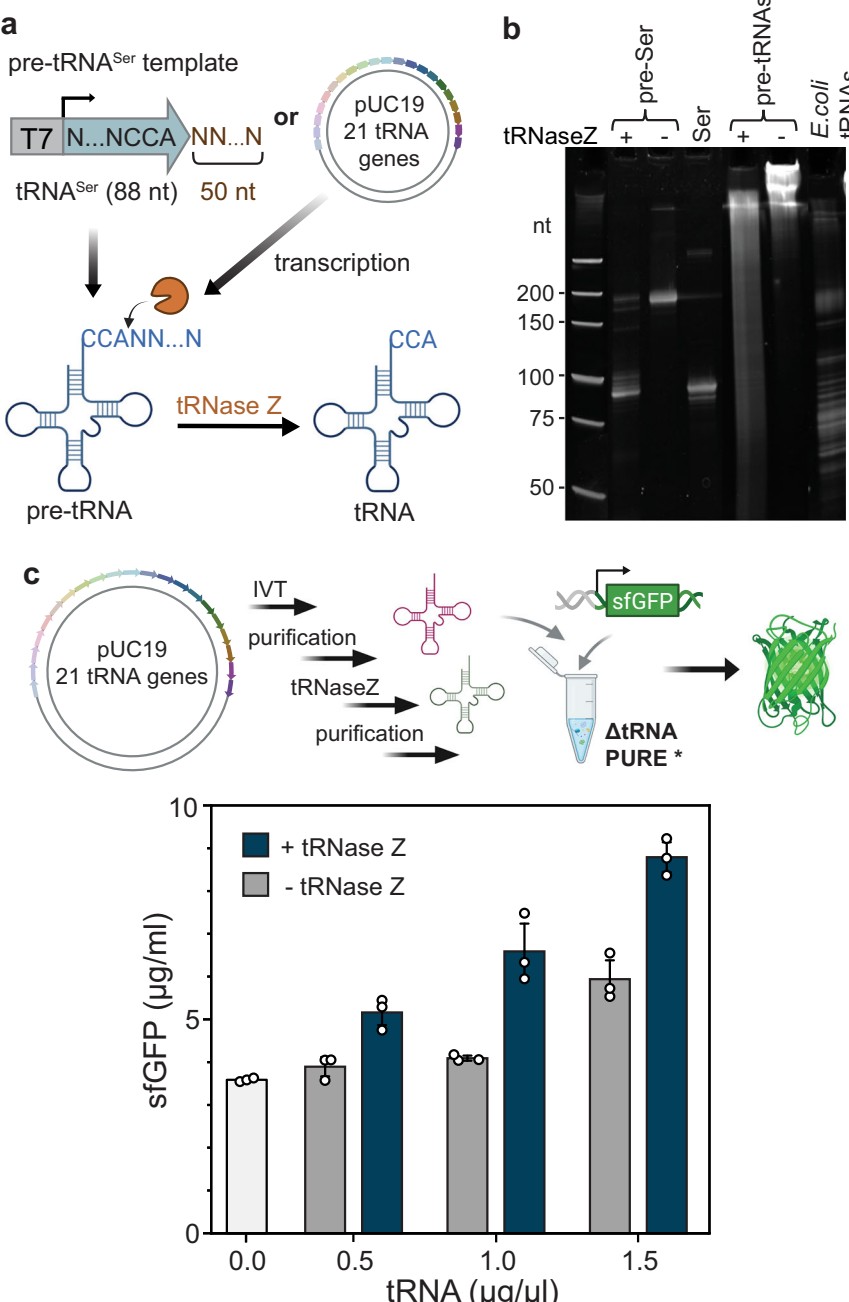

**Fig. 3 | Post-transcriptional cleavage of pre-tRNA with _T. maritima_ tRNase Z. a** A schematic of using _T. maritima_ tRNase Z to trim the _E. coli_ pre-tRNA and production of mature tRNA with CCA triple at the 3' end. Created in BioRender. Maerkl, S. (2025) https://BioRender.com/eh7ck4h. **b** UREA-PAGE analysis of products treated with _T. maritima_ tRNase Z. **c** Activity test of _T.m_ tRNase Z-treated pre-tRNAs in tRNA PURE system (_n_ = 3). Each dot represents a data point from an independent replicate. The schematic above the plot was created in BioRender. Maerkl, S. (2025) https://BioRender.com/eh7ck4h. Bars and error bars represent the mean and standard deviation. Source data are provided as a Source Data file.

E6840S), while the negative control rings were devoid of tRNAs. We observed steady-state fluorescence with the uniform IVT tRNAs, resulting in a steady-state sfGFP concentration of 3.5 µg/ml. Although this sfGFP level was much lower than the sfGFP concentration of 90 µg/ml achieved with E. coli tRNAs, this result confirmed the viability of IVT tRNAs for further experiments. Stable fluorescence levels of the mScarlet tracer validated the experimental fidelity and functionality of all chemostats (Supplementary Fig. 6).

Next, we attempted continuous in situ transcription of tRNAs from various concentrations of ltDNAs or nicked plasmid with 4 nM sfGFP template (Fig. 4e, f). Positive control rings were continuously supplied with 1.5 µg/µl commercial E. coli tRNAs for experiments with

ltDNAs and 0.5 µg/µl uniform IVT tRNAs for experiments with nicked plasmid. For both ltDNA and nicked plasmid conditions we observed an initial peak in sfGFP concentration, up to 5 µg/ml, similar to that obtained when using IVT tRNAs. Unfortunately, this initial peak was followed by a sharp decay into complete loss of synthesis activity, indicating that while the system was able to transcribe sufficient tRNAs for initial sfGFP synthesis, the amount of tRNA continuously synthesized wasn't able to support sfGFP synthesis in a sustainable fashion.

This could be due to the concentration of the tRNA template being lower than required, due to tRNA synthesis dominating mRNA synthesis or vice versa, or due to excessive resource loading of the system at 4 nM sfGFP template concentration, which could all result in

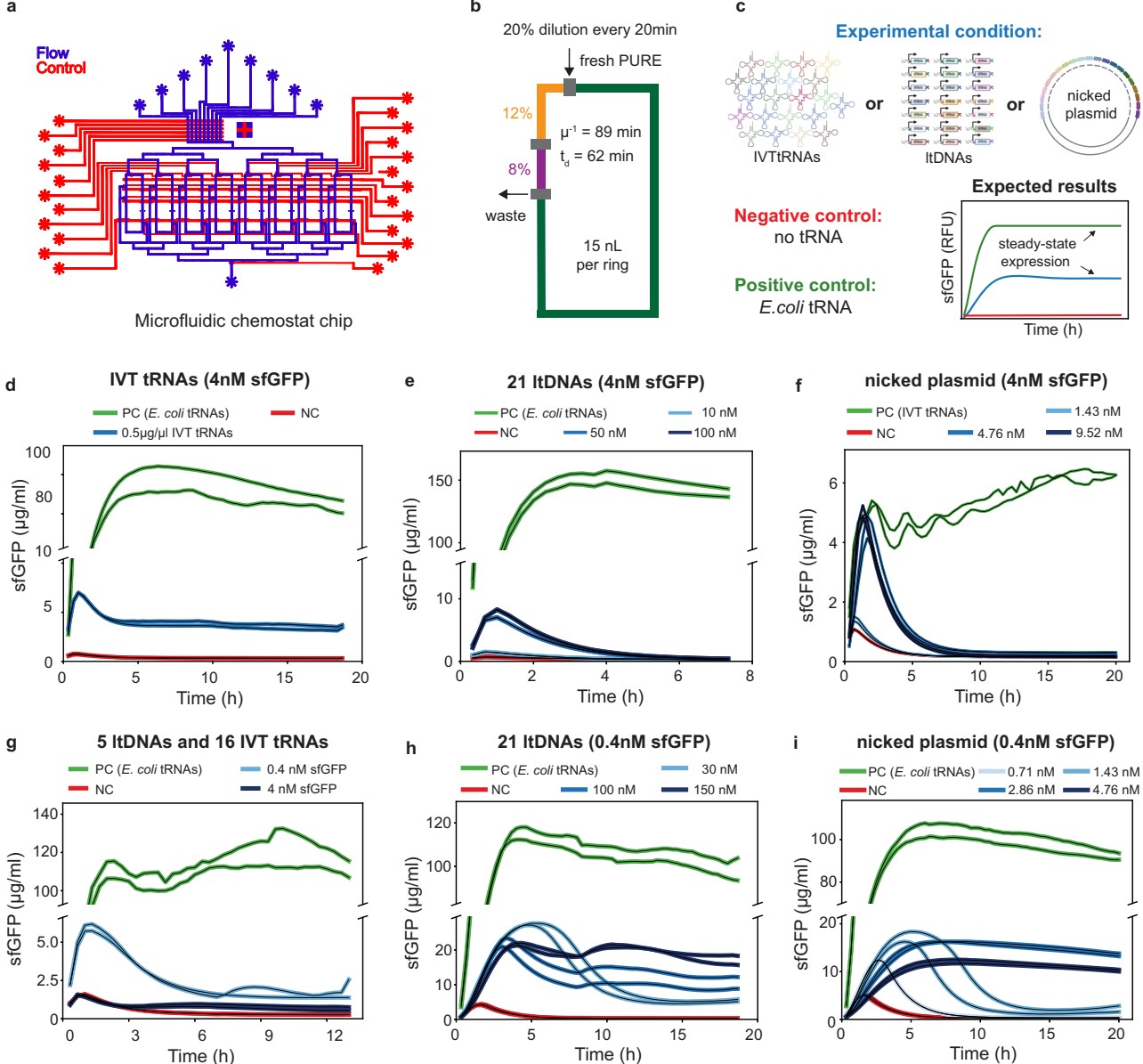

**Fig. 4 | Steady-state protein expression coupled with in situ tRNA synthesis in a microfluidic chemostat. a** Design schematic of the microfluidic chemostat used, which was slightly modified from ref. 9. The schematic shows the flow layer (blue) and the control layer (red) of the chip, with 8 chemostat rings and various fluidic inlets. **b** In each chemostat ring, 20% of the reaction volume was diluted out every 20 min. Dilution rate $\mu = -\ln(C_t/C_0) \cdot t^{-1}$, residence time $\mu^{-1}$ and dilution time $t_d = \ln(2) \, \mu^{-1}$. Chemostat initialization and dilution protocols are described in detail in Supplementary Tables 5 and 6. Each dilution cycle consists of two steps: first loading the protein/ribosome mix (red) through the 20% dilution fraction followed by loading the DNA/energy mix (green) through the 12% dilution fraction, thereby maintaining the desired 3:2 ratio of DNA/energy mix to protein/ribosome mix. **c** The various experimental conditions studied have been described here alongside a model graph for expected results. In the following graphs, the positive controls of the experiments are in green while the negative controls are in red. The schematic of Experimental condition was created in BioRender. Maerkl, S. (2025) https://BioRender.com/eh7ck4h. **d** Continuous protein synthesis achieved in the chemostat using purified IVT tRNAs from ltDNAs (*E. coli* tRNAs used as PC) ($n = 2$). **e, f** In situ synthesis of tRNAs from ltDNAs (**e**, *E. coli* tRNAs used as PC) or nicked plasmid (**f**, IVT tRNAs used as PC) at various input concentrations and a sfGFP template concentration of 4 nM ($n = 2$). **g** In situ synthesis of five tRNAs from ltDNAs with 16 recombinant IVT tRNAs added to the reaction. A sfGFP template concentration of 4 and 0.4 nM was tested (*E. coli* tRNAs used as PC) ($n = 2$). In situ synthesis of all 21 tRNAs from ltDNAs (**h**) or nicked plasmid (**i**) at various concentrations and a sfGFP template concentration of 0.4 nM (*E. coli* tRNAs used as PC) ($n = 2$). Source data are provided as a Source Data file.

a subsequent decay in sfGFP synthesis. The first hypothesis could be excluded because the two highest DNA input concentrations tested for both linear template and plasmid DNA resulted in similar initial peak heights, suggesting that similar amounts of tRNAs were initially generated and being indicative of an upper limit in the amount of tRNAs generated in PURE. This consistent observation with both the plasmid and ltDNAs suggested a possible resource competition problem in continuous reaction conditions requiring both mRNA and in situ tRNA

transcription. To address resource competition in mRNA vs tRNA synthesis, we attempted to run the reactions with higher concentration of NTPs, but only observed a minor improvement at 1.25 × the standard concentration of NTPs with no steady state obtained (Supplementary Fig. 7). The reaction completely died at 2 × NTP concentrations, eliminating this as a possible solution.

To test the resource-loading hypothesis, we decided to titrate sfGFP template concentration. In situ tRNA transcription was

performed in benchtop batch reactions using 0.4 and 4 nM sfGFP templates. Higher sfGFP yield was consistently observed with 0.4 nM sfGFP template at all tested tRNA template concentrations (Supplementary Fig. 8), suggesting that the amount of tRNAs synthesized in situ is perhaps insufficient for protein synthesis from large quantities of mRNA generated with 4 nM sfGFP template. The addition of 8 nM nicked plasmid to reactions containing 0.4 nM sfGFP template and a saturated concentration of *E. coli* tRNA (4 µg/µL) resulted in a reduced sfGFP yield (Supplementary Fig. 9), indicating possible resource competition, either between tRNA and mRNA synthesis or between transcription and translation processes. Based on these results, 0.4 nM sfGFP template was tested in subsequent chemostat reactions. In all the following chemostat experiments, positive control rings were supplied with 1.5 µg/µl *E. coli* tRNAs. We first attempted to in situ transcribe only five of 21 tRNAs from ltDNAs (23.8 nM of 5 ltDNAs) while also titrating sfGFP template concentration at 4 nM and 0.4 nM (Fig. 4g). While the rings with 4 nM sfGFP template closely followed the trends of the negative controls, we observed a higher peak in sfGFP concentration in the rings with 0.4 nM sfGFP template. Furthermore, these peaks subsequently decayed into a low, but non-zero, steady state of 1.5 µg/ml sfGFP concentration, which was very promising and suggested that switching to 0.4 nM sfGFP template concentration could be a possible solution to the resource loading problem.

In line with these observations, we tested the in situ transcription of all 21 tRNAs from ltDNA at 30, 100, and 150 nM input concentrations with 0.4 nM sfGFP template concentration on the microfluidic chemostat (Fig. 4h). At 30 nM ltDNA input concentration, we observed a peak in sfGFP concentration up to 27 µg/ml, followed by a decay into a steady state of 5 µg/ml sfGFP. By contrast, higher stable steady states of 10.5 µg/ml and 17 µg/ml sfGFP were obtained at 100 nM and 150 nM ltDNA input concentrations, respectively, achieving continuous in situ transcription of a full tRNA set alongside sustained long-term protein synthesis. Following successful transcription of tRNAs from ltDNAs, we also tested different concentrations of nicked plasmid template in conjunction with 0.4 nM sfGFP template. Steady-state sfGFP expression was observed with 1.43, 2.86 and 4.76 nM plasmid but not with 0.71 nM plasmid (Fig. 4i). The highest steady-state sfGFP concentration of 13.6 µg/ml was obtained with 2.86 nM nicked plasmid, suggesting an increased competition for resources between tRNA synthesis and mRNA synthesis within the range of 2.86–4.76 nM plasmid concentration, leading to reduction of sfGFP synthesis. By genetically encoding fluorescent aptamers into sfGFP mRNA and tRNA$^{Leu}$, we were able to quantify mRNA and tRNA synthesis levels. Higher concentrations of tRNA template resulted in a decrease in mRNA and a concurrent increase in tRNA, quantifying the direct competition between mRNA and tRNA synthesis (Supplementary Fig. 11). These data highlighted the importance of carefully balancing the various template concentrations for achieving sustained tRNA transcription and protein expression.

## Discussion

We reconstituted a complete set of 21 IVT tRNAs for cell-free protein synthesis by optimizing the tRNA sequence of tRNA$^{Asn}$ and tRNA$^{fMet}$, as well as improving the promoter for tRNA$^{Ile}$ and tRNA$^{Trp}$. Our approach eliminates the need for additional 5′ processing enzymes to generate mature tRNAs. Whereas previous work required the exogenous addition of several chemically synthesized tRNAs[19], we were able to achieve protein expression in the PURE system by in situ transcribing the full set of 21 tRNAs from linear DNA templates or a single nicked plasmid in standard batch reactions. By further optimizing reaction conditions, particularly by reducing resource loading through excessive reporter DNA template, we were ultimately able to perform concurrent in situ tRNA synthesis and protein expression using microfluidic chemostats, achieving long-term steady-state reactions. This is an important step towards the realization of an autopoietic biochemical system and synthetic cell.

Although the use of a nicked plasmid encoding all 21 tRNAs was successful in generating functional tRNAs, the need for a nicked plasmid is also sub-optimal. While it is in principle possible to combine the use of a nicking enzyme to generate a nicked plasmid with DNA replication, we explored an alternative strategy that more closely resembles the naturally occurring tRNA maturation processes based on post-transcriptional 3′ processing. We employed the 3′ processing enzyme tRNase Z from *T. maritima* to remove excess nucleotides from the 3′ end of precursor tRNAs. Cleavage was highly effective for a single test tRNA, but also occurred when challenged with a highly heterogeneous pre-tRNA mixture transcribed from an un-nicked plasmid template. This process resulted in functional tRNAs supporting protein synthesis in the PURE system. This is thus a promising approach for generating mature tRNAs from plasmid-encoded tRNA genes, but additional work is required to optimize this approach and it remains to be assessed whether tRNase Z can be directly integrated into the PURE system.

A recent study by Miyachi et al. reported the use of RNase P, self-cleaving ribozyme (HDVR), and T4 PNK to produce mature and functional tRNAs in vitro[40]. Miyachi et al. also demonstrated that RNase P can cleave two directly linked tRNAs and produce two functional tRNAs without the need for a specific recognition sequence, possibly by recognizing a tRNA structure. While both systems demonstrated the feasibility of cell-free protein synthesis with in situ transcribed tRNAs, the highest protein synthesis yields achieved with 21 IVT tRNAs were lower than those achieved with *E. coli* tRNAs. Miyachi et al. reported 3.7 µg/ml luciferase synthesized with tRNA array or IVT tRNAs in their system. In comparison, we obtained 35 µg/ml sfGFP synthesized with weighted IVT tRNAs, 29 µg/ml sfGFP with ltDNAs, 34 µg/ml sfGFP with nicked plasmid, and 8 µg/ml sfGFP with tRNase Z-treated tRNAs in batch reactions. On chemostats, we achieved steady-state concentrations of 17 µg/ml sfGFP using ltDNAs and 13.5 µg/ml sfGFP using nicked plasmid (Fig. 4h, i), these steady levels represent 17.2% and 14.7% synthesis capacity, respectively, compared to steady-state levels of their positive control reactions using E. coli tRNAs. The synthesis capacity in chemostats could be further improved by the addition of dialysis chambers for continuous exchange of energy resources[11], which would help in alleviating the resource competition between tRNA and mRNA synthesis.

Our results highlighted the potential of balancing the abundance of IVT tRNAs for improved protein yield (Fig. 1e, f) and fine-tuning tRNA stoichiometry during in situ synthesis could therefore be beneficial. It is possible to use T7 promoters of varying strength to further tune the expression levels of each individual tRNA gene[45]. Enhancing protein synthesis requires not only optimizing IVT but also improving translational processes, particularly the interactions between IVT tRNAs and key translational components such as aminoacyl-tRNA synthetases, mRNA, EF-Tu, and the ribosome. While some PTMs are reported to be essential in mediating these interactions, the function of many others is unknown and awaits further investigation. Given the minimal nature of the PURE system, the IVT tRNA-based PURE system provides a flexible platform for studying the functional roles of specific PTMs in translation by supplementing the system with identified tRNA-modifying enzymes or chemically modified tRNAs. The insights obtained from these studies can, in turn, help improve cell-free protein synthesis with IVT tRNAs. Although few studies have focused on protein translation with IVT tRNAs, extensive efforts have been dedicated to incorporating non-canonical amino acids and developing tRNA-based therapeutics, where translation efficiency has been enhanced through the engineering of tRNAs and their associated translational components[46–50]. Building on insights from these studies, it may be possible to further enhance protein synthesis in the PURE system with IVT tRNAs in the

future. Aside from the PURE system, in situ tRNA synthesis could also be conducted in lysate systems, which could eliminate the need for adding tRNAs to the reaction[51].

The ability to in situ transcribe a full set of tRNAs in the PURE system will promote applications such as genetic code expansion and genetic code reprogramming[30,40,52–54] in addition to laying a critical foundation for future progress in synthetic cell engineering. The successful coupling of in situ transcription of 21 tRNAs with protein expression in a batch reaction using the PURE system, along with long-term continuous steady-state in situ transcription of all 21 tRNAs and protein expression in microfluidic chemostats, marks a significant advancement towards constructing a fully self-regenerating biochemical system. While many challenges remain, rapid advances in synthetic biology are bringing us closer to the realization of a fully autopoietic biochemical system.

## Methods

### Preparation of IVTtRNA from linear and circular templates

The 21 tRNAs were selected based on a previous study[31]. Plasmids containing a single tRNA gene were obtained from Dr. Yoshihiro Shimizu. These plasmids were used as templates in a PCR to generate linear tRNA templates (ltDNAs) for transcription. The sequence of primers for PCR reactions and ltDNAs are provided in the Supporting Information (Supplementary Tables 1 and 2). A circular template containing all 21 tRNA genes, referred to as pUC19_21 tRNA genes (GenScript), was synthesized by inserting 21 tRNA genes into a pUC19 vector with a T7 promoter upstream and an Nt.BspQI site downstream of each tRNA gene (Supplementary Fig. 2). Another circular template containing all 21 tRNA genes, referred to as pETA_21 tRNA genes (GenScript), was synthesized by inserting 21 tRNA genes into a pET-A vector with a T7 promoter upstream and an Nt.BspQI site as well as a T7 terminator downstream of each tRNA gene (Supplementary Fig. 4). To prepare the nicked plasmid, plasmid pUC19_21 was incubated with Nt.BspQI (NEB) at 50 °C for 2 h, followed by purification using the DNA Clean & Concentrator kit (Zymo). IVT reaction was performed with HiScribe T7 Quick High Yield RNA Synthesis Kit (NEB) at 37 °C for 3 h. For each 20 µl IVT reaction, 0.2 µg of purified ltDNA or 0.5 µg of purified Nt.BspQI nicked plasmid was used. The transcribed tRNA (IVT tRNA) was purified using the Monarch RNA Cleanup Kit (NEB), eluted in nuclease-free water and stored at −80 °C until use. tRNA and DNA concentrations were quantified with NanoDrop based on A260. The size and purity of tRNA was analyzed with 15% Mini-PROTEAN TBE-Urea Gels (Biorad) and SYBR Gold (Invitrogen) staining.

### Preparation of tRNase Z

The gene for *T. maritima* tRNase Z with 6xHis-tag at the C-terminus (Supplementary Table 3) was cloned into pET-15b vector between XbaI and NdeI restriction sites (GenScript). The plasmid pET-15b_*T. maritima*_His-tRNase Z was transformed into BL21 (DE3) pLysS cells for protein expression. Cells were grown in TB media at 37 °C under shaking until an OD600 of around 0.6 was reached. Cells were then induced with 0.5 mM IPTG at 37 °C and harvested after 2 h. Cells were resuspended in lysis buffer containing 20 mM HEPES, pH 7.5, 700 mM NaCl, 25 mM imidazole, and lysed via sonication (Vibra cell 75186, probe tip diameter: 6 mm, 11 cycles of 20 s ON pulse and 20 s OFF pulse, 70% amplitude). The cells were pelleted by centrifugation at 20,000 × *g* for 20 min at 4 °C, and the supernatant was purified with Ni-NTA affinity chromatography. The proteins were washed with buffer containing 20 mM HEPES, pH 7.5, 700 mM NaCl, 60 mM imidazole and eluted in buffer containing 20 mM HEPES, pH 7.5, 700 mM NaCl, 250 mM imidazole. The eluted fractions were pooled and dialyzed for 16 h at 4 °C in dialysis buffer containing 150 mM NaCl, 20 mM HEPES, pH 7.5. Protein was then mixed with 20% glycerol and stored at −80 °C. Protein quality was analyzed by 4–20% Mini-PROTEIN TGX Precast Protein Gels (Bio-Rad).

### tRNase Z cleavage assay

Pre-tRNA[Ser] containing extra nucleotides after the 3′-CCA was used as a substrate to test the activity of our purified tRNase Z. The template sequence of the pre-tRNA[Ser] is provided in the Supporting Information (Supplementary Table 1). For a 10 µl reaction, we mixed 5 µl of pre-tRNA[Ser] (concentration varies from 2 to 8 µg/µl), 2 µl of 5× reaction buffer (100 mM HEPES pH 7.5, 25 mM MgCl2, 1 M KCl) and 3 µl of 0.64 mg/ml tRNase Z. After incubation at 37 °C for 1 h, the pre-tRNA with and without tRNase Z treatment were analyzed with 15% Mini-PROTEAN TBE-Urea Gels (Biorad) and SYBR Gold (Invitrogen) staining. The same reaction was performed for the pre-tRNAs transcribed from the plasmid pUC19_21 tRNA genes. To test the functionality of the tRNase Z-digested pre-tRNAs for cell-free protein expression, the product was further purified using the Monarch RNA Cleanup Kit (NEB) and eluted in nuclease-free water.

### Preparation of energy solution omitting tRNA

Energy solution omitting tRNA (ES ΔtRNA) was prepared as described previously with slight modifications[55]. The components for 4 × ESΔtRNA are 1.2 mM of each amino acid, 47.2 mM magnesium acetate, 400 mM potassium glutamate, 8 mM ATP and GTP, 4 mM CTP, UTP, and TCEP, 80 mM creatine phosphate, 0.08 mM folinic acid, 8 mM spermidine, and 200 mM HEPES, pH 7.5.

### Protein expression

DNA templates for sfGFP and mCherry expression were designed according to the reduced codon table and were synthesized (Supplementary Tables 3 and 4). In vitro protein synthesis reactions (10 µl) were prepared by mixing the indicated concentration of protein expressing template, the indicated concentration of tRNA or tRNA template with the reagents supplied in the PURExpress Δ(aa, tRNA) Kit (NEB, E6840S): 2 µl Solution A (minus aa, tRNA), 3 µl Solution B, and 1 µl amino acid master mix, and brought to a final volume of 10 µl with addition of nuclease free water. As a positive control, the *E.coli* tRNA from the PURExpress Δ(aa, tRNA) Kit (NEB, E6840S) was used. The reaction mixtures were loaded to a black/clear bottom, 384-well microtiter plates (Corning) and incubated at 30 °C at constant shaking for 16 h and measured on a SynergyMX platereader (BioTek) every 4 min. For sfGFP: excitation 485 nm, emission 515 nm, 70% gain. For mCherry: excitation 580 nm, emission 620 nm, 100% gain. To test the activity of tRNase Z-treated tRNA product in PURE system, Solution A (minus aa, tRNA) and amino acid master mix were replaced with laboratory-made energy solution. The PURE system with Solution A (minus aa, tRNA) and amino acid master mix from NEB Kit was referred to as ΔtRNA PURE, whereas the one with laboratory-made ESΔtRNA was referred to as ΔtRNA PURE*. All sfGFP RFU was converted to absolute concentration by a calibration curve (Supplementary Fig. 10a) obtained with purified sfGFP protein expressed with PUREfrex®2.1 kit. NEBExpress®Ni-NTA Magnetic Beads (NEB, S1423S) were used for purification of his-tagged sfGFP protein. The concentration of purified sfGFP protein was determined with the Bradford assay.

### Microfluidic chemostat design and setup

The microfluidic device was fabricated as previously described by Lavickova et al.[9] Silicon molds for the flow and control layer were fabricated by standard photolithography techniques, using AZ 10XT-60 as the photoresist for the flow layer mold and SU-8 for the control layer mold. A chrome mask for the flow layer was patterned using a VPG200 (Heidelberg Instruments). For the flow layer mold, a silicon wafer was primed with HMDS, spin-coated with AZ 10XT-60 to a height of 15 µm and soft-baked on a Süss ACS 200 GEN3. The baked wafer was exposed with the chrome mask on a Süss MA6Gen3 for two steps of 18 s each with a wait period of 10 s between them. The exposed wafer was developed on the Süss ACS 200 GEN3 and baked at 160 °C for 2 h. The silicon wafer mold for the control layer was first primed with an oxygen

plasma etch (TePla 300) for 7 min. SU-8 3025 (Kayaku) was spin-coated on to the primed wafer to obtain a height of 40 μm and baked at 95 °C for 13 min. The control layer was directly patterned onto the wafer using a VPG200 (Heidelberg Instruments), followed by post-exposure bake and development with propylene glycol monomethyl ether acetate. Finally, the wafer was hard baked at 135 °C for 2 h. Both wafers were treated with trimethylchlorosilane overnight, prior to soft lithography, to facilitate PDMS removal from them. A PDMS mixture with 5:1 ratio of elastomer to curing agent was used for the flow layer, while a mixture of 20:1 ratio was spin-coated onto the control layer mold. Both PDMS coated wafers were baked for 20 min in an oven set at 80 °C, following which the flow layer PDMS chips were cutout and aligned to the control layers. After subsequent baking for an hour at 80 °C, the aligned chips were punched and plasma bonded to a glass slide.

For long-term cell-free experiments, a fabricated microfluidic chip was first primed by filling the control channels with de-ionized water. The flow channels were primed with 10 mM Tris-HCl buffer, also used as the wash buffer between dilution steps, followed by flowing 2% BSA solution through for 10 min to prevent unwanted protein adsorption on the channel walls. Next, the channels were rinsed with buffer once again before loading the PURE reaction components prepared for each reactor ring in sequence. The DNA/energy mix were loaded in one FEP tubing, while the protein/ribosome mix was separated into another FEP tubing to avoid prior mixing of components before entering the rings. These two-component solutions were loaded into each ring in a 3:2 volume ratio, respectively. A peristaltic pump was operated at 1.5 Hz to mix components within rings. Laboratory-made ESΔtRNA and Solution B (protein/ribosome mix) PURExpress Δ(aa, tRNA) Kit (NEB) were used for reactions in microfluidic devices. In all experiments, a 1.67× DNA/energy mix was prepared by mixing the laboratory-made ESΔtRNA with sfGFP template and either tRNA or tRNA templates based on experiment parameters. Each 1.67× DNA-energy mix was also supplemented with TCEP (final concentration: 4 μM), 4.17 μM (final concentration: 2.5 μM) Chi DNA and RNAse inhibitor (final concentration: 2 U/μL). A 2.5× protein/ribosome mix was prepared by mixing Solution B with TCEP, RNAse inhibitor (final concentration: 2 U/μL) and mScarlet protein as tracer for visualization. The entire setup was enclosed at 34 °C in a chamber. Fluorescence in all reactor rings was tracked over time using 20× magnification on an automated inverted microscope. The solenoid valves and the microscope were operated by a custom LabVIEW and MATLAB program. The mScarlet tracer was visualized using a mCherry filter, and GFP synthesis was monitored via a FITC filter. RFU values were measured from images using ImageJ by using a consistent rectangular bounding box and manually shifting it to the center of the fluidic channel for deriving the median RFU value inside the box. Resulting RFU values were converted to absolute concentration by a calibration curve (Supplementary Fig. 10b, c) obtained with purified sfGFP protein expressed with PUREfrex®2.1 kit and plotted with Python. Each line graph in the plots represents the mScarlet tracer (Supplementary Fig. 6) or sfGFP concentration (Fig. 4) in one ring over the course of an experiment.

## Aptamer assay

To quantify sfGFP, mRNA, and tRNA synthesis in one reaction, we attached aptamers Pepper[56] and Clivia[57] to mRNA and tRNA, respectively. The dyes, HBC620 (Targetmol) and NBSI574 (FR Biotechnology), were used for imaging Pepper and Clivia aptamers, respectively. The DNA templates for sfGFP-Pepper and tRNA^Leu-Clivia were synthesized by Twist Bioscience (Supplementary Table 3). Transcripts of mRNA-pepper and tRNA-Clivia were prepared through IVT with HiScribe T7 Quick High Yield RNA Synthesis Kit (NEB), followed by purification using Monarch RNA Cleanup Kit (NEB). RNA concentration was quantified with NanoDrop based on A260. To test the orthogonality of aptamers and dyes, 1 μM aptamer was incubated with and 4 μM dye. Emission spectra was collected after excitation at 490 or 580 nm

(Supplementary Fig. 11a, b). The stability of the dye and aptamers was tested by monitoring the fluorescence at 30 °C for 10 h (Supplementary Fig. 11c, d). RFU of m/tRNA-aptamers was converted to absoute concentrations by calibration curves (Supplementary Fig. 11e, f). The cell-free protein expression reactions (10 μl) were prepared by mixing 3 μM HBC620, 3 μM NBSI574, 0.4 nM sfGFP-Pepper template, the indicated concentration of 21 ltDNAs and tRNA^Leu-Clivia with the reagents supplied in the PURExpress Δ(aa, tRNA) Kit (NEB): 2 μl Solution A (minus aa, tRNA), 3 μl Solution B, and 1 μl amino acid master mix, and brought to a final volume of 10 μl with addition of nuclease free water. We note that the concentration of tRNA^Leu-Clivia template was adjusted to the same level of individual ltDNA in each reaction. The reaction mixtures were loaded to a black/clear bottom, 384-well microtiter plates (Corning) and incubated at 30 °C at constant shaking for 16 h and measured on a SynergyMX platereader (BioTek) every 4 min. For sfGFP: excitation 485 nm, emission 515 nm, 70% gain. For Pepper-HBC620: excitation 580 nm, emission 620 nm, 100% gain. For Clivia-NBSI574: excitation 490 nm, emission 580 nm, 100% gain.

## Statistics and reproducibility

No statistical method was used to predetermine sample size. A small fraction of data from microfluidic experiments was excluded if a malfunction of the chip occurred due to delamination or precipitation in a given chemostat. The experiments were not randomized. The Investigators were not blinded to allocation during experiments and outcome assessment.

## Reporting summary

Further information on research design is available in the Nature Portfolio Reporting Summary linked to this article.

## Data availability

The data generated in this study are provided in the Supplementary Information and Source Data file. Source Data file and microfluidic mask designs are provided with this paper https://doi.org/10.5281/zenodo.15657846[58]. Source data are provided with this paper.

## Code availability

Data acquisition and microscopy for the chemostat experiments were performed with LabVIEW and Matlab code, available at https://doi.org/10.5281/zenodo.15658120[59].

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

## Acknowledgements

The authors would like to express their gratitude to Dr. Kithmie Malagoda Pathiranage, Dr. Rigumula Wu, Dr. Ru Zheng, Dr. Kelvin Lau, Dr. Yoan Duhoo, Dr. Maria Lopez Malo, Dr. Laura Grasemann, Dr. Laura Roset, Ragunathan Bava Ganesh, Pao-Wan Lee, and Matthis Guillaume Lugnier for helpful input and comments. This work was supported by a Swiss National Science Foundation Spark Grant (220737, F.L.) and a Swiss National Science Foundation MINT grant (214843, S.J.M).

## Author contributions

F.L. prepared tRNAs and performed related biochemical experiments. A.K.B. performed chemostat experiments. F.L., A.K.B., and S.J.M. designed experiments, analyzed data, and wrote the manuscript.

## Competing interests

The authors declare no competing interests.
