## [Transparent Peer Review file · Nature Communications]

Continuous in situ synthesis of a complete set of tRNAs sustains steady-state translation in a recombinant cell-free system

Corresponding Author: Professor Sebastian Maerkl

Version 0:

Reviewer comments:

Reviewer #1

(Remarks to the Author)

In this manuscript, the authors report a 21-tRNA system capable of powering cell-free protein synthesis (CFPS) by the PURE CFPS system. They demonstrate the realization of this pool of tRNAs for protein synthesis from DNA templates. They demonstrate the possibility of getting these tRNA synthesized in situ using two methods, either from a nicked plasmid or using tRNase Z digestion if synthesized from a non-nicked plasmid. Finally, they demonstrate the feasibility of a continuous in situ synthesis of tRNA in a PURE cell-free system, achieving steady-state tRNA and protein synthesis in a microfluidic chemostat system.

Overall, this work is potentially interesting for the CFPS community, especially users of the PURE system, as it addresses cell-free synthesis of tRNA, a key challenge of cell-free bottom-up synthetic biological systems. That said, the position of the work with respect to previously published achievements is not clear. In particular, the work presented in this manuscript does not seem to be very innovative when compared to the work published by Hibi and coworkers in 2020. The manuscript lacks data critical to understanding the biochemical mechanisms involved in the synthesis of tRNA and the optimization of CFPS, which remains low compared to adding *E. coli* tRNAs. The presentation of the work is not optimum, changes should be made to improve the quality of the manuscript.

Major comments:

- 1) Many figures (especially Fig1 f&h, Fig2 c&d, Fig3c and Fig4 b,c,d,e,f&g) present fluorescence data in Relative Units (RFU). How is it possible to do that in 2025? To ensure the transferability and comparison of the data obtained with other setups and outside of the lab, an absolute quantification must be performed and the data presented must be converted into an absolute unit (such as sfGFP true concentration).
- 2) A major limitation of the work presented here is the lack of efficacy of CFPS achieved with the tRNA synthesis approaches when compared to the titers reached for the control using commercial *E. coli* tRNA. Even if potential explanations and solutions are somewhat discussed (such as post-transcription modification of the tRNAs), this is not satisfactory. Additional experiments should be performed to evaluate potential solutions to mitigate that gap, especially post-transcription modifications of the tRNAs to determine how that step is critical to improve the approach. In the paper by Hibi and coworkers in 2020, the question of the effect of post-transcription modification was studied for several tRNAs to show that they have important effects on CFPS yields.
- 3) The evaluation of the resource competition aspect of tRNA in situ synthesis is not strong enough. To further evaluate these aspects, an experiment measuring the decrease of sfGFP synthesis in the presence of both a saturating concentration of commercial tRNA and the presence of the tRNA expressing plasmids tested could be done, or any similar experiments to clarify this point.
- 4) In the first figure of the paper, the authors demonstrate a strong impact of tRNA abundance distribution on the efficiency of the mixture to promote CFPS. However, when it comes to experiments involving in situ tRNA synthesis, the lack of quantification of the abundance level of individual tRNAs makes it impossible to measure this effect. The authors need to find a way of quantifying the distribution of functional tRNAs synthesized by their plasmids to assess and discuss these effects.
- 5) Still on the effects of tRNA synthesized distribution effect, in the case of *T. maritima* processed tRNA, there may be a major impact related to the amount of long transcript containing multiple tRNAs. This effect could be investigated and

optimized by changing the architecture of the template plasmid, reducing the number of T7 promoters, adding terminators, and shuffling the position of tRNA encoding cassettes in the plasmid. The paper would greatly benefit from additional data exploring some of these possibilities.

Minor comment:

- Figure 1f could be improved to make it more readable, right now there are points in the graph that are hard to see.
- In the current state of the paper, the source (provider) of the E. coli tRNA mix used as a control in figure 1 and others was not specified. This information must be provided in the material and methods section.

Reviewer #2

(Remarks to the Author)

In this manuscript, Li et al present an important step toward self-sustaining cell-free in vitro systems by achieving continuous in situ synthesis of a full set of 21 tRNAs in the PURE TX-TL system. The authors improve the transcriptional yield of several weakly expressed tRNAs, demonstrate functional protein synthesis using linear DNA templates or a single plasmid, and extend the system to steady-state protein production in microfluidic chemostats. The combination of biochemical engineering and microfluidic implementation is well executed and the results are relevant for the bottom-up synthetic biology community.

Major comments:

1. Line 166, The authors write: "Upon comparing the efficiency of ltdNAs with nicked plasmid at equal total gene concentrations (168 nM total ltdNA concentration is equivalent to 8 nM nicked plasmid), we found that uniform ltdNAs have higher activity than the nicked plasmid."

However, in the microfluidic chemostat experiment performed with 4 nM GFP plasmid, 150 nM ltdNA performed more than 10 times worse than 9.52 nM nicked plasmid (judging by peak height of GFP expression, Figure 4c vs. 4d). This is quite a discrepancy compared to the batch experiments. How do the authors account for this difference?

2. Line 236, The authors write: "Unfortunately this initial peak was followed by a sharp decay into complete loss of synthesis activity, indicating that while the system was able to transcribe enough tRNAs for initial sfGFP synthesis, the amount of tRNA continuously synthesized wasn't able to support sfGFP synthesis in a sustainable fashion."

When the authors tried to increase yields by increasing the concentration of NTP, did they make sure that sufficient amounts of supplemental Mg²⁺ were also supplied?

3. In Figure 3c, in the minus tRNA^{AsZ} reactions, there appears to be a significant increase in activity from 1 ug/uL to 1.5 ug/uL. This is quite interesting. A similar weak trend can be seen in Figure S3. Can the authors speculate why translation is occurring even though the pre-tRNAs have not been processed by RNase Z? Or is there weak background activity by residual E. coli tRNA processing enzymes in the PURE system?

4. Figure 4b: What is the authors' interpretation of why there is also an initial peak for the IVTtRNA control? Is this also typical for conventional PURE reactions with standard E. coli tRNAs?

5. The use of *T. maritima* tRNAse Z is a clever workaround to avoid nicked plasmids. However, the relatively low cleavage efficiency warrants further investigation. Could optimization of flanking sequences or intermediate terminators improve maturation?

6. The authors may wish to comment on how their in situ tRNA synthesis strategy could be applied beyond PURE (e.g. in lysate-based systems). This would increase the broader relevance of the paper.

Minor comments:

1. No spaces between number and unit in several occasions.
2. The phrase "hallmarks of live" (line 2 of the introduction) should be corrected to "hallmarks of life."
3. The authors might also want to include this recent preprint in their discussion:
<https://www.biorxiv.org/content/10.1101/2025.02.15.638384v1>

Reviewer #3

(Remarks to the Author)

Li and colleagues described transcription of all tRNAs essential for canonical translation in vitro: the full set of tRNA for each amino acids, and initiation methionine. This is a well written paper, providing elegant solution to one of the most pressing problems in the process of engineering synthetic living cell from self-replicating in vitro reagents.

Notably, the authors correctly recognize that the work presented here is not a first demonstration of simultaneous transcription and utilization of all of those tRNAs for in vitro translation, citing previous work on 15 tRNAs. This doesn't diminish the novelty of this work, the authors still demonstrated in situ synthesis of 6 previously unsolved tRNA, making a first truly complete demonstration of full complement of tRNA needed for translation.

Two biggest drawbacks that diminish utility of this work are lack of stoichiometry control, and lack of post-transcriptional modifications.

Post-transcriptional modifications are a key element necessary for tRNA activity. Authors hypothesize that the lack of those modifications might be the main reason responsible for lower yields of protein expression.

Similarly, authors note that it's possible tRNAs are not available at the ratios that are optimal for the templates, codon optimized for bacterial differences in tRNA abundance.

While those two issues require separate significant efforts to solve, it would be valuable for the readers, and to facilitate utilization of those results in future work, to provide some discussion about possible ways in which the platform presented in this paper can be further developed to introduce those two things. The ability to control ratios of tRNA, and post-transcriptional modifications, will be crucial for use if the final goal of this paper, and the whole field, is to be realized.

Previous work in this field, including papers cited by the Authors (so they are familiar with that work) used RNaseP. Yet here the Authors chose to use RNase Z. This choice, while proven correct by the results, is not sufficiently explained in the discussion. More reason for using an enzyme different than previous work would help the reader understand the choice, and better understand the improvements of this method over previously used one.

The use of the chemostat is unclear. What was the point of using a chemostat, over a batch reaction, in this case?

While the advantages of the chemostat have been elegantly demonstrated in previous papers, it is not clear in this work what was the advantage of microfluidic chip over bulk solution synthesis to demonstrate tRNA production and use in translation.

For data shown on Fig S7, what controls were done to be sure that aptamer folding and t/mRNA folding didn't interfere with one another?

More translation (which would be expected with more tRNA) of a message (any reporter protein) would result in more remodeling by the ribosome. It would be useful to discuss, and experimentally interrogate, how would this impact aptamer folding?

It would be useful to interrogate the influence of intrinsic terminators and abortive cycling on the efficiency and abundance of particular tRNAs. Since the final yields of the tRNA are the biggest limiting factor in this system, minimizing abortive cycling might be one important way to improve the results.

The no-tRNA controls, expression in delta tRNA PURE, shows measurable eGFP signal (for example figure 1f and h). Is is autofluorescence of the reaction mix, or is this PURE not truly devoid of tRNA? We know from previously published work, and this reviewer's own experience, that Δ tRNA PURE produces no measurable translation, but GFP indeed has a lot of autofluorescence. Perhaps a single control with a reporter with a better background, using luminescence or other enzymatic readout instead of fluorescent protein, could help to demonstrate that starting PURE is truly devoid of endogenous tRNA.

Version 1:

Reviewer comments:

Reviewer #1

(Remarks to the Author)

In the revised version of their manuscript, the authors have addressed the issues pointed out in the first round. They performed useful additional experiments and expanded the discussion section to better explain the perspective of their work, particularly in terms of hypotheses to address the central issue of low expression observed in in situ-synthesized tRNA. Overall, the manuscript seems suitable for publication after a few minor edits.

Minor comment: figure S9 seems to contain a typo in the x-axis caption, there are 2 bars labeled -nicked plasmid -E. coli tRNA.

Reviewer #2

(Remarks to the Author)

The authors satisfactorily answered my open questions.

Reviewer #3

(Remarks to the Author)

The authors adequately addressed all my questions and comments. I have no more complaints. I think it's a good paper and it will be valuable to the community.

Response to reviewers - Continuous *in situ* synthesis of a complete set of tRNAs sustains steady-state translation in a recombinant cell-free system

Fanjun Li, Amogh Kumar Baranwal, and Sebastian J. Maerkl

Institute of Bioengineering, School of Engineering, École Polytechnique Fédérale de Lausanne, Lausanne, Switzerland

May 27, 2025

We thank all reviewers for their insightful and helpful comments in improving this manuscript. In the revised manuscript, we have addressed the issues raised by the reviewers, with major changes highlighted. Our point-by-point responses to all reviewers' comments are listed below.

Reviewer 1

In this manuscript, the authors report a 21-tRNA system capable of powering cell-free protein synthesis (CFPS) by the PURE CFPS system. They demonstrate the realization of this pool of tRNAs for protein synthesis from DNA templates. They demonstrate the possibility of getting these tRNA synthesized *in situ* using two methods, either from a nicked plasmid or using tRNase Z digestion if synthesized from a non-nicked plasmid. Finally, they demonstrate the feasibility of a continuous *in situ* synthesis of tRNA in a PURE cell-free system, achieving steady-state tRNA and protein synthesis in a microfluidic chemostat system.

Overall, this work is potentially interesting for the CFPS community, especially users of the PURE system, as it addresses cell-free synthesis of tRNA, a key challenge of cell-free bottom-up synthetic biological systems. That said, the position of the work with respect to previously published achievements is not clear. In particular, the work presented in this manuscript does not seem to be very innovative when compared to the work published by Hibi and coworkers in 2020. The manuscript lacks data critical to understanding the biochemical mechanisms involved in the synthesis of tRNA and the optimization of CFPS, which remains low compared to adding *E. coli* tRNAs. The presentation of the work is not optimum, changes should be made to improve the quality of the manuscript.

Major points

- **Q1:** Many figures (especially Fig 1 f & h, Fig 2c & d, Fig 3c and Fig 4b, c, d, e, f & g) present fluorescence data in Relative Units (RFU). How is it possible to do that in 2025? To ensure the transferability and comparison of the data obtained with other setups and outside of the lab, an absolute quantification must be performed and the data presented must be converted into an absolute unit (such as sfGFP true concentration).

Response: We agree with the reviewer that absolute, quantitative values are always superior to arbitrary units. We do want to point out that two recent manuscripts on tRNA and AARS synthesis / regeneration in the PURE system also use arbitrary units (<https://www.biorxiv.org/content/10.1101/2025.02.15.638384v1>, <https://www.science.org/doi/10.1126/sciadv.adt6269>). Nonetheless, since we feel the same as the reviewer about this issue we've converted sfGFP RFU to absolute concentration and updated all relevant figures. We also updated our results and discussion sections accordingly. The calibration curves for sfGFP measurements in plate reader and chemostat are now included as Figure S10 and described in the material and methods section.

- **Q2:** A major limitation of the work presented here is the lack of efficacy of CFPS achieved with the tRNA synthesis approaches when compared to the titers reached for the control using commercial *E. coli* tRNA. Even if potential explanations and solutions are somewhat discussed (such as post-transcription modification of the tRNAs), this is not satisfactory. Additional experiments should be performed to evaluate potential solutions to mitigate that gap, especially post-transcription modifications of the tRNAs to determine how that step is critical to improve the approach. In the paper by Hibi and coworkers in 2020, the question of the effect of post-transcription modification was studied for several tRNAs to show that they have important effects on CFPS yields.

Response: We agree with the reviewer that the current protein yields achieved with *in situ* synthesized tRNAs are considerably lower than when using purified *E. coli* tRNAs. This is the case for all related published work in this area as far as we know. But, our work represents one of the first demonstrations, together with the Miyachi et al. paper which was published nearly concurrently with our paper, that it is at all possible to *in vitro* transcribe all 21 tRNAs, which in turn sets the foundation for future optimization and improvements as the reviewer pointed out.

Since we generated quantitative sfGFP reporter values in response to Q1 above, we can also now report that our approach is consistently better than what was described by Miyachi et al. (<https://www.biorxiv.org/content/10.1101/2025.02.15.638384v1>) who reported a synthesis capacity of 3.7 $\mu\text{g}/\text{ml}$ luciferase whereas our approach returned between 8 and 34 $\mu\text{g}/\text{ml}$ of sfGFP depending on the specific system used. These values correspond to 300-1300nM of sfGFP which also compares favorably with what was reported

in Hibi et al. (<https://www.nature.com/articles/s42003-020-1074-2>). More importantly, in our chemostat experiments we achieved a steady-state protein synthesis level that was as high as 17% compared to a positive control reaction using purified *E. coli* tRNAs, which shows that under these advanced conditions use of *in situ* transcribed tRNAs is in fact not too far away from optimality.

We concur that integrating post-transcriptional modifications could potentially enhance CFPS yields and provide valuable insights, pursuing such experiments would require extensive additional work (as noted by Reviewer 3) and is beyond the scope of the present study. We therefore expanded our discussion to outline how the platform presented in this manuscript could be further developed to address the current low protein yield.

- **Q3:** The evaluation of the resource competition aspect of tRNA *in situ* synthesis is not strong enough. To further evaluate these aspects, an experiment measuring the decrease of sfGFP synthesis in the presence of both a saturating concentration of commercial tRNA and the presence of the tRNA expressing plasmids tested could be done, or any similar experiments to clarify this point.

Response: We performed an additional test as advised by the reviewer and observed a decrease of sfGFP synthesis in the presence of both a saturating concentration of commercial tRNA and the tRNA expressing plasmid, supporting the resource competition hypothesis. The result is now shown as Figure S9. We updated the main text as follows:

" *In situ* tRNA transcription was performed in benchtop batch reactions using 0.4 nM and 4 nM sfGFP templates. Higher sfGFP yield was consistently observed with 0.4 nM sfGFP template at all tested tRNA template concentrations (Figure S8), suggesting that the amount of tRNAs synthesized *in situ* is perhaps insufficient for protein synthesis from large quantities of mRNA generated with 4 nM sfGFP template. The addition of 8 nM nicked plasmid to reactions containing 0.4 nM sfGFP template and a saturated concentration of *E. coli* tRNA (4 $\mu\text{g}/\mu\text{L}$) resulted in a reduced sfGFP yield (Figure 1), indicating possible resource competition, either between tRNA and mRNA synthesis or between transcription and translation processes. Based on these results, 0.4 nM sfGFP template was tested in subsequent chemostat reactions."

Figure 1: Figure S9

- **Q4:** In the first figure of the paper, the authors demonstrate a strong impact of tRNA abundance distribution on the efficiency of the mixture to promote CFPS. However, when it comes to experiments involving in situ tRNA synthesis, the lack of quantification of the abundance level of individual tRNAs makes it impossible to measure this effect. The authors need to find a way of quantifying the distribution of functional tRNAs synthesized by their plasmids to assess and discuss these effects.

Response: We appreciate the reviewer’s insightful comment. Recognizing the limitations of denaturing PAGE and qPCR in resolving RNAs of similar sizes and sequences, we have considered advanced sequencing-based methods such as Nano-tRNAseq, mim-tRNAseq, AQRNA-seq, and OTTR. However, given that impurities in the tRNA pool may differ by only a few nucleotides, particularly at the 3’ end, even current sequencing techniques face challenges in achieving accurate profiling. As such, the quantification of individual tRNA species falls outside the scope of this study.

- **Q5:** Still on the effects of tRNA synthesized distribution effect, in the case of *T. maritima* processed tRNA, there may be a major impact related to the amount of long transcript containing multiple tRNAs. This effect could be investigated and optimized by changing the architecture of the template plasmid, reducing the number of T7 promoters, adding terminators, and shuffling the position of tRNA encoding cassettes in the plasmid. The paper would greatly benefit from additional data exploring some of these possibilities.

Response: We agree with the reviewer that the listed variables are worth exploring. While a systematic evaluation of the suggested variables is beyond the scope of the current study,

we tested a plasmid design with a wild type T7 terminator inserted after each tRNA gene. However, not much improvement was observed with the new design, which could be due to the insufficient termination of wild type T7 terminator. Stronger T7 terminators could be tested next (<https://doi.org/10.1093/g3journal/jkac070>). In future work, we aim to investigate these aspects more thoroughly to optimize template design for improved tRNA yield and translation efficiency. We now included this data as Figure S5 (below) and updated the discussion section.

Figure 2: Figure S5

Main text: "To improve the maturation process, we tested another circular plasmid where a wild-type T7 terminator was inserted downstream of each tRNA gene. Transcripts from this plasmid were incubated with tRNase Z and subsequently tested for activity in the Δ tRNA PURE system, but no significant improvement was observed (Figure S5). This could be due to the insufficient termination efficiency of the wild-type T7 terminator, resulting in the persistence of long transcripts. To address this, stronger T7 terminators (<https://doi.org/10.1093/g3journal/jkac070>) could be evaluated next. Besides, shuffling the positions of tRNA genes (<https://www.biorxiv.org/content/10.1101/2025.02.15.638384v1>) may offer further enhancements."

Minor points

- **Q1:** Figure 1f could be improved to make it more readable, right now there are points in the graph that are hard to see.

Response: We revised Figure 1f accordingly:

Figure 3: Figure 1f

- **Q2:** In the current state of the paper, the source (provider) of the *E. coli* tRNA mix used as a control in figure 1 and others was not specified. This information must be provided in the material and methods section.

Response: The *E. coli* tRNA used in this study was from the PURExpress Δ (aa, tRNA) Kit (NEB, E6840S). We've added this information in main text and in the material and methods section (SI document).

Main text: "...We added the sfGFP template (4 nM) and various concentrations of uniform IVT tRNAs or *E. coli* tRNAs (NEB, E6840S) to the Δ tRNA PURE system..."

Material and Methods: "As a positive control, the *E. coli* tRNA from the PURExpress Δ (aa, tRNA) Kit (NEB, E6840S) was used."

Reviewer 2

In this manuscript, Li et al present an important step toward self-sustaining cell-free in vitro systems by achieving continuous in situ synthesis of a full set of 21 tRNAs in the PURE TX-TL system. The authors improve the transcriptional yield of several weakly expressed tRNAs, demonstrate

functional protein synthesis using linear DNA templates or a single plasmid, and extend the system to steady-state protein production in microfluidic chemostats. The combination of biochemical engineering and microfluidic implementation is well executed and the results are relevant for the bottom-up synthetic biology community.

Major points

- **Q1:** Line 166, The authors write: "Upon comparing the efficiency of ltdDNAs with nicked plasmid at equal total gene concentrations (168 nM total ltdDNA concentration is equivalent to 8 nM nicked plasmid), we found that uniform ltdDNAs have higher activity than the nicked plasmid." However, in the microfluidic chemostat experiment performed with 4 nM GFP plasmid, 150 nM ltdDNA performed more than 10 times worse than 9.52 nM nicked plasmid (judging by peak height of GFP expression, Figure 4c vs. 4d). This is quite a discrepancy compared to the batch experiments. How do the authors account for this difference?

Response: Following verification of their functionality in Fig 4b, IVT tRNAs were used as the positive control (mentioned in the graph legend) in Figure 4d vs commercial tRNAs used as positive controls in the rest of the experiments. We specifically chose IVT tRNAs as the positive control here to highlight the similarity in the peak expression values observed when using IVT tRNAs vs the tRNA plasmid template. In reality, 150 nM ltdDNA performed similarly to 9.52 nM nicked plasmid but they appear different due to using IVT tRNAs as the positive control for Fig 4d and using RFU as the units for y-axis. As seen in Fig 4b, sfGFP expression from IVT tRNAs is much lower compared to commercial tRNAs. This is now improved for clarity as we've switched to using absolute concentration values for y-axis in all figures following a suggestion from Reviewer 1. It's immediately visible after this change that 150 nM ltdDNAs performed similarly to 9.52 nM nicked plasmid based on the peak sfGFP concentrations recorded in both cases.

- **Q2:** Line 236, The authors write: "Unfortunately this initial peak was followed by a sharp decay into complete loss of synthesis activity, indicating that while the system was able to transcribe enough tRNAs for initial sfGFP synthesis, the amount of tRNA continuously synthesized wasn't able to support sfGFP synthesis in a sustainable fashion." When the authors tried to increase yields by increasing the concentration of NTP, did they make sure that sufficient amounts of supplemental Mg²⁺ were also supplied?

Response: We performed an initial condition screening on a plate reader before testing on chemostats and observed that elevated concentrations of Mg²⁺ adversely affected sfGFP synthesis, but we did not titrate NTP together with various concentrations of Mg²⁺. Given the critical role of Mg²⁺ in many aspects of transcription and translation, and its interac-

tion with NTPs, we acknowledge that our current data on NTP titration alone may not be sufficient, more comprehensive titration of NTPs at different concentration of Mg²⁺ might improve protein synthesis.

- **Q3:** In Figure 3c, in the minus tRNaseZ reactions, there appears to be a significant increase in activity from 1 ug/ul to 1.5 ug/ul. This is quite interesting. A similar weak trend can be seen in Figure S3. Can the authors speculate why translation is occurring even though the pre-tRNAs have not been processed by tRNase Z? Or is there weak background activity by residual E. coli tRNA processing enzymes in the PURE system?

Response: It might be due to residual endogenous tRNA processing enzymes (contaminants) in the PURE system. It could also be due to the increased levels of short tRNA segments being produced by T7 RNAP during abortive cycles that happened to be mature and functional tRNAs. We now include this discussion in the main text as follow:

“Interestingly, a significant increase in activity of untreated pre-tRNAs from 1 ug/ul to 1.5 ug/ul was observed, which could be due to the residual E. coli tRNA processing enzymes in the PURE system, or due to increased levels of short tRNA segments produced by T7 RNAP during abortive cycles that happen to be mature and functional tRNAs.”

- **Q4:** Figure 4b: What is the authors’ interpretation of why there is also an initial peak for the IVT tRNA control? Is this also typical for conventional PURE reactions with standard E. coli tRNAs?

Response: It might be that 0.5 $\mu\text{g}\cdot\mu\text{l}^{-1}$ of IVT tRNA was at the lower threshold of tRNA required to sustain a steady state in the chemostat, thereby resulting in an initial peak. It is possible that this is linked to the resource loading that we observed. In this particular case it could be that as mRNA levels rise from the start of the reaction that there is a short period of optimal mRNA concentration giving rise to a peak. Once mRNA levels exceed this optimal level, a decrease is observed due to loading effects (too few tRNAs and too much mRNA). In our previous work on protein self-regeneration using E. coli tRNA we also observed initial peaks, as well as peaks during wash-out phases (<https://doi.org/10.1038/s41467-020-20180-6>). In fact it was these wash-out phase peaks that indicated to us that T7 RNAP concentration was likely not optimal in the system for these previous experiments.

- **Q5:** The use of T. maritima tRNase Z is a clever workaround to avoid nicked plasmids. However, the relatively low cleavage efficiency warrants further investigation. Could optimization of flanking sequences or intermediate terminators improve maturation?

Response: To optimize tRNA maturation, we recently tested a plasmid with a wild type T7 terminator inserted after each tRNA gene. However, not much improvement was observed. We now included this data as Figure S5 (also attached to the response to reviewer 1's Q5) and updated the discussion in main text. It could be due to the insufficient termination of wild type T7 terminator and plenty of long transcripts remained. There is certainly much more optimization work to be done, such as optimizing the plasmid design (try stronger T7 terminators and other variables as suggested by reviewer 1) and engineering tRNase Z to improve the maturation process.

- **Q6:** The authors may wish to comment on how their *in situ* tRNA synthesis strategy could be applied beyond PURE (e.g. in lysate-based systems). This would increase the broader relevance of the paper.

Response: We appreciate the reviewer's suggestion and have added the following lines at the end of the discussion on PTMs and engineering tRNA abundances:
Aside from the PURE system, *in situ* tRNA synthesis could also be conducted in lysate systems, which could eliminate the need for adding tRNAs to the reaction.

Minor points

- **Q1:** No spaces between number and unit in several occasions.

Response: We have corrected all such instances of inappropriate spacing.

- **Q2:** The phrase "hallmarks of live" (line 2 of the introduction) should be corrected to "hallmarks of life."

Response: We have changed "live" to "life".

- **Q3:** The authors might also want to include this recent preprint in their discussion - <https://www.biorxiv.org/content/10.1101/2025.02.15.638384v1>

Response: This is clearly a highly relevant manuscript to include. As it was published only 3 days prior to us submitting to biorxiv we were not yet aware of it to include it in our biorxiv submission. We've updated our manuscript by discussing the pre-print in several places. For example, regarding the 3'-end processing of tRNA, the optimization of template design,

and comparing the protein synthesis efficiency of the two systems.

Reviewer 3

Li and colleagues described transcription of all tRNAs essential for canonical translation *in vitro*: the full set of tRNA for each amino acids, and initiation methionine. This is a well written paper, providing elegant solution to one of the most pressing problems in the process of engineering synthetic living cell from self-replicating *in vitro* reagents.

Notably, the authors correctly recognize that the work presented here is not a first demonstration of simultaneous transcription and utilization of all of those tRNAs for *in vitro* translation, citing previous work on 15 tRNAs. This doesn't diminish the novelty of this work, the authors still demonstrated *in situ* synthesis of 6 previously unsolved tRNA, making a first truly complete demonstration of full complement of tRNA needed for translation.

Two biggest drawbacks that diminish utility of this work are lack of stoichiometry control, and lack of post-transcriptional modifications.

- **Q1:** Post-transcriptional modifications are a key element necessary for tRNA activity. Authors hypothesize that the lack of those modifications might be the main reason responsible for lower yields of protein expression. Similarly, authors note that it's possible tRNAs are not available at the ratios that are optimal for the templates, codon optimized for bacterial differences in tRNA abundance. While those two issues require separate significant efforts to solve, it would be valuable for the readers, and to facilitate utilization of those results in future work, to provide some discussion about possible ways in which the platform presented in this paper can be further developed to introduce those two things. The ability to control ratios of tRNA, and post-transcriptional modifications, will be crucial for use if the final goal of this paper, and the whole field, is to be realized.

Response: We appreciate the reviewer's comments and have added a paragraph on this topic in the discussion section as follows:

Our results highlighted the potential of balancing the abundance of IVT tRNAs for improved protein yield (Figure 1e, f) and fine-tuning tRNA stoichiometry during *in situ* synthesis could therefore be beneficial. It is possible to use T7 promoters of varying strength to further tune the expression levels of each individual tRNA gene. Enhancing protein synthesis requires not only optimizing IVT, but also improving translational processes, particularly the interactions between IVT tRNAs and key translational components such as aminoacyl-tRNA synthetases, mRNA, EF-Tu, and the ribosome. While some PTMs are reported to be essential in mediating these interactions, the function of many others are unknown and await further investigation. Given the minimal nature of the PURE system, the IVT tRNA-based

PURE system provides a flexible platform for studying the functional roles of specific PTMs in translation by supplementing the system with identified tRNA-modifying enzymes or chemically modified tRNAs. The insights obtained from these studies can, in turn, help improve cell-free protein synthesis with IVT tRNAs. Although few studies have focused on protein translation with IVT tRNAs, extensive efforts have been dedicated to incorporating non-canonical amino acids and developing tRNA-based therapeutics, where translation efficiency has been enhanced through the engineering of tRNAs and its associated translational components. Building on insights from these studies, it may be possible to further enhance protein synthesis in the PURE system with IVT tRNAs in the future.

- **Q2:** Previous work in this field, including papers cited by the Authors (so they are familiar with that work) used RNaseP. Yet here the Authors chose to use RNase Z. This choice, while proven correct by the results, is not sufficiently explained in the discussion. More reason for using an enzyme different than previous work would help the reader understand the choice, and better understand the improvements of this method over previously used one.

Response: RNase P is a well-established enzyme for 5'-end processing of tRNA. In contrast, maturation of the 3'-end is more challenging. One proposed strategy involves incorporating a self-cleaving ribozyme at the 3'-end of the tRNA, which generates a tRNA bearing a 2',3'-cyclic phosphate. This intermediate is then converted to a mature 3'-OH end through enzymatic dephosphorylation, enabling aminoacylation and participation in translation. Coincidentally, while we were preparing our manuscript, this method was experimentally validated by Miyachi et al. (<https://www.biorxiv.org/content/10.1101/2025.02.15.638384v1>). To streamline 3'-end processing *in vitro*, we aimed to develop a single-enzyme solution that performs the complete maturation. Therefore, we tested the one step maturation with tRNase Z. We have updated the main text as follows:

"One possible strategy for 3'-end maturation involves incorporating a self-cleaving ribozyme at the 3'-end of the tRNA, which generates a tRNA bearing a 2',3'-cyclic phosphate. This intermediate can then be converted to a mature 3'-OH end through enzymatic dephosphorylation, enabling aminoacylation and participation in translation. Coincidentally, while we were preparing our manuscript, this method was experimentally validated by Miyachi et al. To streamline 3'-end processing *in vitro*, we aimed to develop a single-enzyme solution that performs the complete maturation."

- **Q3:** The use of the chemostat is unclear. What was the point of using a chemostat, over a batch reaction, in this case? While the advantages of the chemostat have been elegantly demonstrated in previous papers, it is not clear in this work what was the advantage of microfluidic chip over bulk solution synthesis to demonstrate tRNA production and use in translation.

Response: The chemostat allows us to test long-term steady state in-situ tRNA synthesis alongside translation in the cell-free system. It also provides deeper insight into the observed resource competition between continuous tRNA synthesis and translation, wherein we observed complete cessation of protein synthesis under heavy loading, something not readily apparent in batch reactions.

- **Q4:** For data shown on Fig S7, what controls were done to be sure that aptamer folding and t/mRNA folding didn't interfere with one another?

Response: The aptamers were integrated into a commonly used RNA scaffold, three-way junction F30, which can promote proper folding by providing a stable structural framework. We assumed the aptamer's folding and functionality will not be significantly affected by m/tRNA as this method has been widely used. We didn't perform additional comparison with aptamer alone. Instead, we verified the stability of the m/tRNA-aptamer fusions and their associated dyes by monitoring the fluorescence of the aptamer-dye complex at a constant concentration over several hours. We also verified the orthogonality of the aptamers and the dyes. For the influence of aptamer, we observed an increase in background sfGFP protein level with sfGFP-pepper template compared with sfGFP only template, but we are not sure whether it is due to the effect of pepper aptamer on mRNA structure or other possibilities. However, since the purpose of this assay was to compare m/tRNA and protein yield across varying tRNA template concentrations, we believe this background effect does not significantly alter the overall trends in m/tRNA dynamics.

- **Q5:** More translation (which would be expected with more tRNA) of a message (any reporter protein) would result in more remodeling by the ribosome. It would be useful to discuss, and experimentally interrogate, how would this impact aptamer folding?

Response: The aptamer was appended to the mRNA after the stop codon and spaced with several nucleotides. We anticipate that upon encountering the stop codon, the ribosome will dissociate from the mRNA, thereby not interfering with the folding of the aptamer.

- **Q6:** It would be useful to interrogate the influence of intrinsic terminators and abortive cycling on the efficiency and abundance of particular tRNAs. Since the final yields of the tRNA are the biggest limiting factor in this system, minimizing abortive cycling might be one important way to improve the results.

Response: We appreciate the reviewer's suggestion and have considered this aspect in our work. Specifically, we tested a T7 RNAP mutant, previously reported to reduce abortive

transcription (<https://doi.org/10.1038/s41587-022-01525-6>). However, no noticeable improvement was observed in our hand. As a result, we did not pursue this approach further at this stage, but we agree that minimizing abortive cycling remains a promising avenue for future optimization.

- **Q7:** The no-tRNA controls, expression in delta tRNA PURE, shows measurable eGFP signal (for example figure 1f and h). Is it autofluorescence of the reaction mix, or is this PURE not truly devoid of tRNA? We know from previously published work, and this reviewer's own experience, that Δ tRNA PURE produces no measurable translation, but GFP indeed has a lot of autofluorescence. Perhaps a single control with a reporter with a better background, using luminescence or other enzymatic readout instead of fluorescent protein, could help to demonstrate that starting PURE is truly devoid of endogenous tRNA.

Response: We tested the autofluorescence of the reaction mix (Δ tRNA PURE) without adding sfGFP template, the fluorescence signal of which is lower than that with sfGFP template added, suggesting that there is residual tRNA contamination in the Δ tRNA PURE kit (NEB, E6840S).

Our work suggests a counterintuitive but strong dependence on input reporter DNA concentration in terms of synthesis efficiency. For example if a Δ tRNA control reaction is performed with relatively high reporter DNA input concentrations it would give rise to low or even un-detectable levels of translation (reporter synthesis) due to resource competition. Whereas, if reactions are performed with low input reporter DNA concentrations, the same reaction may now indeed be able to translate small amounts of detectable reporter protein because resource competition is alleviated. We also know that different PURE systems have different levels of contaminants stemming not only from the non-ribosomal PURE proteins but also from the ribosomes.

The use of luminescence readout is useful due to its higher sensitivity, allowing even very small amounts of expressed enzyme (luciferase for example) to be detected. This is generally a last resort if GFP (or other fluorescent protein) production can't be detected. Fluorescent proteins have the considerable advantage that they can be easily calibrated and represent a direct and easily comparable readout of cell-free systems efficiency. Luminescence-based readouts on the other, because of the enzymatic nature of the process, represent yet another level of signal amplification in the system, which is not related to the core processes of transcription and translation. Luminescence-based readouts may appear to be highly efficient, but in reality what determines the maximum signal intensities is not the enzyme concentration but rather a limiting substrate concentration, which is a big caveat.

Response to reviewers - Continuous *in situ* synthesis of a complete set of tRNAs sustains steady-state translation in a recombinant cell-free system

Fanjun Li, Amogh Kumar Baranwal, and Sebastian J. Maerkl

Institute of Bioengineering, School of Engineering, École Polytechnique Fédérale de Lausanne, Lausanne, Switzerland

June 18, 2025

We thank all reviewers for their insightful and helpful comments in improving this manuscript. In the revised manuscript, we have addressed the issues raised by the reviewers.

Reviewer #1 (Remarks to the Author):

In the revised version of their manuscript, the authors have addressed the issues pointed out in the first round. They performed useful additional experiments and expanded the discussion section to better explain the perspective of their work, particularly in terms of hypotheses to address the central issue of low expression observed in *in situ*-synthesized tRNA. Overall, the manuscript seems suitable for publication after a few minor edits.

Minor comment: figure S9 seems to contain a typo in the x-axis caption, there are 2 bars labeled -nicked plasmid -E. coli tRNA.

Response: We've corrected the typo.

Reviewer #2 (Remarks to the Author):

The authors satisfactorily answered my open questions.

Reviewer #3 (Remarks to the Author):

The authors adequately addressed all my questions and comments. I have no more complaints. I think it's a good paper and it will be valuable to the community.